# Telomerase and alternative lengthening of telomeres coexist in the regenerating zebrafish caudal fins

Elena Martínez-Balsalobre [1,2,3,4,7], Monique Anchelin[1,2,7], David Hernández-Silva[1,5], Maria C Mione [6], Victoriano Mulero [2,3,4], Francisca Alcaraz-Pérez [1,2,3,4✉], Jesús García-Castillo [1,2,4✉] & María L Cayuela [1,2,4,5✉]

## Abstract

Telomeres are essential for chromosome protection and genomic stability, and telomerase function is critical for organ homeostasis. Zebrafish is a useful vertebrate model for understanding cellular and molecular mechanisms of regeneration. The regeneration capacity of the caudal fin of wild-type zebrafish is not affected by repetitive amputation, but the behaviour of telomeres during this process has not yet been studied. Here, we characterize the regeneration process in a telomerase-deficient zebrafish model, and study the regenerative capacity after repetitive amputations at different ages. We find that the regenerative efficiency decreases with aging in all genotypes but telomere length is maintained even in telomerase-deficient fish. Our data indicate that telomere length can be maintained by the regenerating cells through the recombination-mediated Alternative Lengthening of Telomeres (ALT) pathway, which likely supports high rates of cell proliferation during the caudal fin regeneration process.

**Keywords** Zebrafish; Fin Regeneration; TERT; ALT Mechanism
**Subject Categories** DNA Replication, Recombination & Repair; Stem Cells & Regenerative Medicine

## Introduction

Tissue regeneration is an evolutionarily conserved response to injury (Morrison et al, 2006), and while all animals regenerate some of their tissues by physiological turnover, only a few can regenerate appendages. Zebrafish (*Danio rerio*) is one of such organisms, able to regenerate retina, fins, heart, spinal cord and other tissues even in advanced ages (Becker et al, 1997; Poss et al, 2003; Poss et al, 2002; Reimschuessel, 2001; Rowlerson et al, 1997).

Because of its regenerative ability, its simple but relevant anatomy, in vivo imaging capability and genetic advantages, zebrafish have become a useful vertebrate model for understanding the cellular and molecular mechanisms of regeneration (Goldsmith and Jobin, 2012). The caudal fin is the most convenient tissue for regenerative studies due to its easy handling and fast regeneration. Adult zebrafish regenerate their caudal fin within 14 days after amputation (Anchelin et al, 2011; Poss et al, 2003).

Several groups have investigated cells and genetic signalling pathways regulating blastema formation (Wehner and Weidinger, 2015). In addition, the regeneration limit of the zebrafish caudal fin was previously investigated (Azevedo et al, 2011; Shao et al, 2011), however, there are very few studies about the implication of telomeres and telomerase in this process. Moreover, the zebrafish has constitutively abundant telomerase activity in somatic tissues from embryos to aged adults (Anchelin et al, 2011; McChesney et al, 2005). Notably, a study on various tissues from aquatic species, including the zebrafish, suggests that telomerase may be important for tissue renewal and regeneration after injury rather than for overall organism longevity (Elmore et al, 2008). Our previous studies about the behaviour of telomeres and telomerase during the regeneration process revealed a direct relationship between telomerase expression, telomere length and efficiency of tissue regeneration in wild-type zebrafish caudal fin (Anchelin et al, 2011). In addition, characterization of the telomerase-deficient zebrafish suggests that telomerase function is crucial for organ homeostasis in zebrafish (Anchelin et al, 2013; Henriques et al, 2013), as occurs in mouse (Blasco et al, 1997).

The aim of this study was to clarify the role of telomerase during the caudal fin regenerating process. We confirmed the outstanding and almost unlimited caudal fin regeneration capability of zebrafish, which only decreases in aged animals, also in the case of the telomerase-deficient genotype. Moreover, we found the telomere length is maintained even in telomerase-deficient genotypes suggesting the involvement of alternative lengthening of telomeres (ALT) pathway.

[1]Grupo de Telomerasa, Cáncer y Envejecimiento, Hospital Clínico Universitario Virgen de la Arrixaca, 30120 Murcia, Spain. [2]Instituto Murciano de Investigación Biosanitaria-Pascual Parrilla (IMIB-PP), 30120 Murcia, Spain. [3]Departamento de Biología Celular e Histología, Facultad de Biología, Universidad de Murcia, 30100 Murcia, Spain. [4]Centro de Investigación Biomédica en Red de Enfermedades Raras (CIBERER), ISCIII, 30100 Murcia, Spain. [5]Heatlh Faculty, Universidad Católica de Murcia (UCAM), Campus de los Jerónimos, s/n, Guadalupe, 30107 Murcia, Spain. [6]Department of Cellular, Computational and Integrative Biology-CIBIO, University of Trento, Trento, Italy. [7]These authors contributed equally: Elena Martínez-Balsalobre, Monique Anchelin. ✉E-mail: palcaraz@um.es; jesus.garcia@ffis.es; marial.cayuela@carm.es

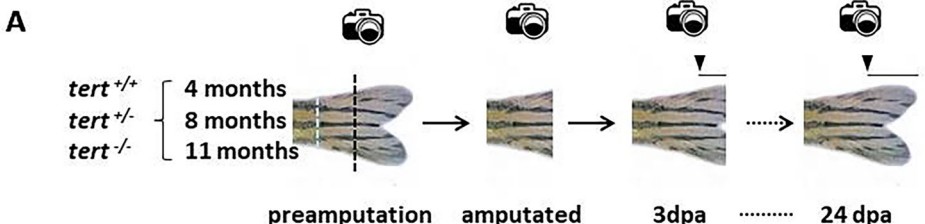

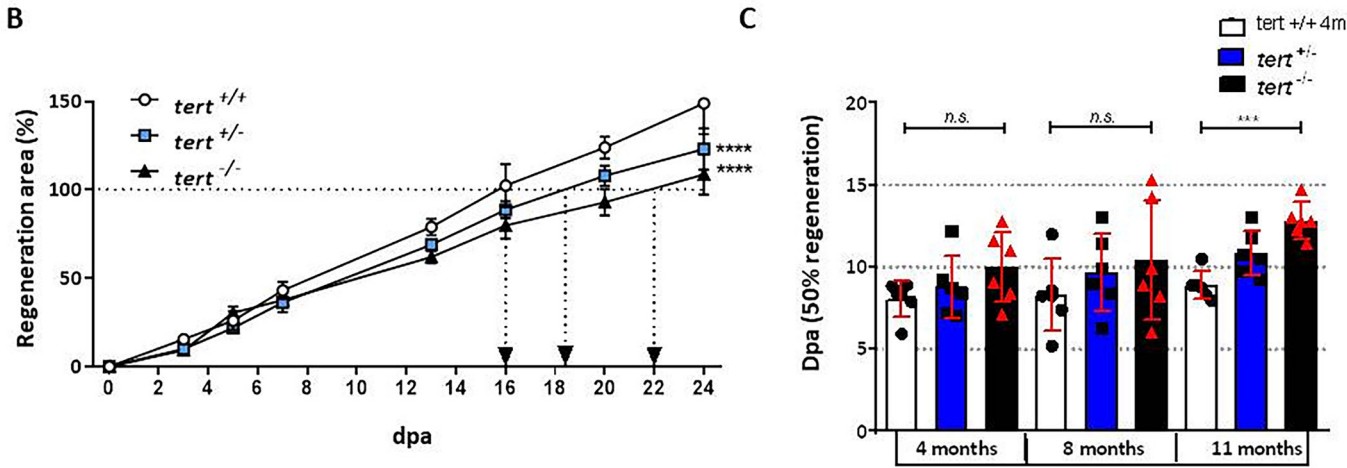

**Figure 1. Fish with different telomerase genotypes regenerate with different efficiency.**

(A) Experimental design of a regeneration assay with a single amputation (black dotted tracing) and growth monitoring (arrowhead and line), at three different ages of life (4, 8, and 11 month old) in tert+/+, tert+/−, and tert−/− zebrafish for 24 days. The caudal peduncle is indicated in a blue dotted line. (B) Caudal fin regeneration slope of 8-month-old zebrafish from the three genotypes. The arrows indicate 100% regeneration of the tail fin. $n = 3$ experiments with two biological replicates each. Data were mean + s.e.m. (C) Days post amputation to reach 50% of the caudal fin regeneration at three different ages and from the three telomerase genotypes. $n = 3$ experiments. Each dot represents a biological replicate. Data were mean + standard deviation ***$P < 0.001$ and ****$P < 0.0001$ for two-way ANOVA plus Dunnett's post hoc test. dpa days post amputation. When exact $p$ values are not indicated is because GraphPad Prism software does not show it. Source data are available online for this figure.

# Results

## Caudal fin regeneration is affected by aging

It is known that after a single excision of the zebrafish caudal fin, a regenerative process is activated, and it takes ~2 weeks to fully regenerate all the tissues and structures that compose a functional fin (Akimenko et al, 2003; Poss et al, 2003). Several studies show that consecutive repeated amputations of zebrafish caudal fin do not reduce its regeneration capacity (Azevedo et al, 2011; Shao et al, 2011); however, changes in telomere length during the regenerative process have not been studied.

To further investigate the role of the catalytic subunit of telomerase (*tert*) during this regenerating process, *tert+/+, tert+/−,* and *tert−/−* (also mentioned later as *tert* mutant or *tert* deficient) specimens from the same genetic background, maintained in the same laboratory conditions, were used for regeneration assays after a single amputation or after repeated amputations.

Zebrafish of all three genotypes and at different life stages (4, 8, and 11 months) were amputated and their regeneration was monitored (Fig. 1A). The fish used showed non-detectable aging phenotype at 4 months old, whereas at 8 and 11 months fish presented the typical *tert* mutant premature aging phenotypes, i.e., backbone curvature, loss of body

mass and hypopigmentation. The 8-month fish regeneration slope (Fig. 1B) showed that *tert+/+* regenerates its entire caudal fin at 16 days post amputation (dpa), faster than *tert+/−* and *tert−/−* zebrafish (at 19 and 22 dpa, respectively). Furthermore, *tert+/−* and *tert−/−* zebrafish need more time to reach 50% of their caudal fin regeneration than wild-type zebrafish at all ages (Fig. 1C), and this difference is significant at 11 months of age, when fish without telomerase are considered to be aged (Anchelin et al, 2013). Thus, all genotypes can regenerate the caudal fin.

Since the ability of caudal fin regeneration in telomerase-deficient zebrafish is not reduced by consecutive clips and the telomere length is maintained after three consecutive clips (Anchelin et al, 2011; Azevedo et al, 2011; Shao et al, 2011), we decided to explore the regeneration process in these fish. We performed a consecutive amputation experiment, cutting the caudal fin every 7 days for 9–12 times, and using "young adult fish", without premature aging phenotype, (4 month old) and "old adult fish", showing aging features, (11 month old) from the three genotypes (Fig. 2A). The resulted regeneration slope showed that the young zebrafish from the three different genotypes were able to regenerate their caudal fins at a similar percentage after 12 consecutive amputations (Fig. 2B). However, in the case of the old adult fish, *tert+/−* and *tert−/−* zebrafish showed a statistically significant reduction of regeneration percentage compared with

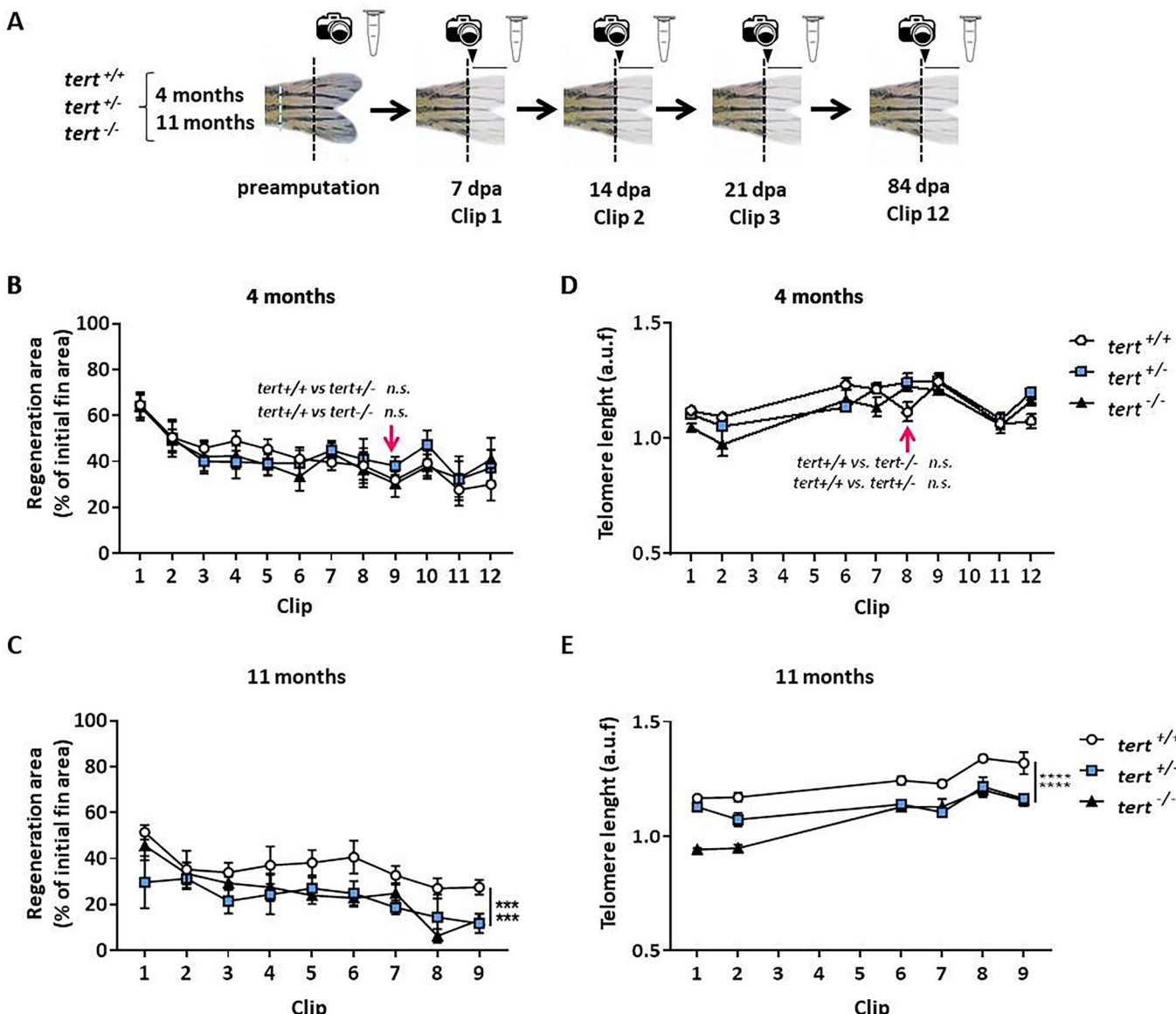

**Figure 2. Telomere length is maintained during consecutive amputations in telomerase-deficient zebrafish.**

(A) Experimental design of a regeneration assay with 12 consecutive amputations every 7 days and growth monitoring (arrowhead and line), in *tert*[+/+], *tert*[+/−], and *tert*[−/−] genotypes, at 4 and 11 months of age. The caudal peduncle is indicated in a blue dotted line. (B) Slopes of the regeneration areas after 12 consecutive clips in the three telomerase genotypes at 4 months of age. *n* = 3 experiments with 2 biological replicates each. (C) Slopes of the regeneration areas after 9 consecutive clips in the three telomerase genotypes at 11 months age. *n* = 3 experiments with two biological replicates each. (D) Measurement of telomere length by flow-FISH of regenerative fin tissue cells in all three telomerase genotypes at 4 months of age *n* = 2 experiments with 1 biological replicate. (E) Measurement of telomere length by Flow-FISH of regenerative fin tissue cells in all three telomerase genotypes at 11 months of age. *n* = 2 experiments with 1 biological replicate. All data were mean + standard deviation. ***$P < 0.001$ and ****$P < 0.0001$ for two-way ANOVA plus Dunnett's post hoc test, calculated using data at clip number 9. When exact *p* values are not indicated is because GraphPad Prism software does not show it. Source data are available online for this figure.

their wild-type *tert*[+/+] counterparts (Fig. 2C), which continues to show a high regeneration percentage, although slower than their younger siblings.

## Telomere length is maintained after consecutive caudal fin amputations

It has already been published and confirmed here, that the regenerative capacity of zebrafish caudal fin is retained even in aged fish. Surprisingly, this regenerative capacity is not abrogated is aged telomerase-deficient fish (*tert*[+/−] and *tert*[−/−]), therefore we decided to assess the dynamics of telomere length. Caudal fin regenerative tissue was collected after each clip during the regeneration assay, and the telomere length was measured by Flow-FISH. Telomere length was maintained in the young zebrafish group even in telomerase-deficient genotypes (Fig. 2D). However, in old zebrafish, the telomere length decreased in *tert*[+/−] and *tert*[−/−] (Fig. 2E), but not as dramatically as expected.

These results suggest that telomerase is not essential for the regeneration of the zebrafish caudal fin and lead us to further investigate the mechanism by which these cells are able to maintain their telomere length throughout repeated amputations, which would translate to a continual renewal of tissues with high cell proliferation rates.

## ALT is involved in telomere length maintenance during caudal fin regeneration

Telomerase is the main telomere maintenance mechanism, but in about 10% of human cancer cells, telomere length is maintained by the alternative lengthening of telomeres (ALT) mechanism (Conomos et al, 2013; Draskovic and Londono Vallejo, 2013). This mechanism is based on homologous recombination and homology-directed telomere synthesis (Pickett and Reddel, 2015). Human ALT cells show a number of unusual characteristics; one of the most striking is an abundance of DNA with telomeric sequences separate from the chromosomes. This extrachromosomal telomeric DNA takes many forms, including predominantly partially single-stranded telomeric circles (referred to as C-circles, as it mostly involves the C-rich strand) (Henson et al, 2009).

Therefore, we set out to detect and quantify circular extrachromosomal telomeric DNA in the cells obtained from the caudal fin regenerative portion using the quantitative and sensitive C-circle assay (CCassay) (Lau et al, 2013) (Fig. 3A). At time zero, there were no significant differences between wild-type and telomerase-deficient fish in C-circles amounts. However, we observed a statistically significant increase above 2 (a value considered indicative of C-circles abundance, (Henson et al, 2009)) of C-circles at 48 h post amputation (hpa) in the cells extracted from the telomerase-deficient zebrafish compared with no changes observed in the cells of the wild-type zebrafish. As controls for C-circle levels, a telomerase cell line (HeLa) and an ALT cell line (SAOS-2) were used (Fig. EV1A).

Other characteristics of ALT cells include highly heterogeneous telomere lengths (ranging from undetectable to extremely long), and rapid changes in telomere length (Cesare and Griffith, 2004). To visualize zebrafish telomeres during regeneration, a quantitative FISH (q-FISH) was performed to calculate relative length of telomere within the blastema interphase cells before amputation and 7 days post-amputations (Fig. 3B–D). We observed a decrease in telomere length after the proliferation process leading to regeneration starts in wild-type zebrafish but not in fish of $tert^{-/-}$ line, where we did not detected changes in the average telomeric length (Fig. 3C). However, we observed a higher telomeric heterogeneity in $tert^{-/-}$ than in $tert^{+/+}$ (Fig. 3D). A higher percentage of short (21.1%) and long telomeres (6.1%) in telomerase-deficient zebrafish was detected at 7 days post amputation compared to the percentage of short (12.1%) and long telomeres (2.6%) found before amputation. However, in $tert^{+/+}$ animals, we observed an increase in long telomeres (9.9%) but a decrease in short telomeres (3.1%) at 7 days post amputation when compared to 0 days post amputation (0.0 and 5.7%, respectively; Fig. 3E).

Since ALT is based on homologous recombination between telomeres, a number of recombination proteins have been shown by genetic analyses to be necessary for telomere maintenance in ALT cells (Martinez et al, 2017), as the MRN complex (MRE11a;

RAD50 and NBS1) (Zhong et al, 2007) and other DNA binding proteins, including the serine/threonine protein kinase, ATR (Collis et al, 2008). On the other hand, chromatin in ALT cells seems to be more relaxed than in telomerase–positive cells. Interestingly, genes encoding ATRX (α-thalassemia/mental retardation syndrome X-linked) and DAXX (death domain-associated protein 6), members of chromatin remodelling complex that are active at telomeres, are strongly associated with ALT and may be involved in ALT suppression, while their loss and/or mutations in these genes is associated with ALT activation (Amorim et al, 2016; Ren et al, 2018; Yost et al, 2019). Therefore, we decided to analyse the mRNA levels of several genes, which may be required to activate or prevent ALT during the regeneration process.

These genes were quantified in the regenerated tissue by qPCR at 24 and 48 hpa. $nbs1$ and $atr$ mRNA levels increased in telomerase-deficient fish at 48 hpa compared to 0 hpa (Fig. 4A,B). The same effect in the expression of these genes was found in wild-type fish regenerating caudal fins. Interestingly, $atrx$ and $daxx$ expression decreased (Fig. 4C,D) at 24 and 48 hpa, in agreement with published data on ALT in cells (Amorim et al, 2016; Ren et al, 2018; Yost et al, 2019), whereas the expression of these genes was unaltered in wild-type animals upon amputation.

To further show that the ALT process is directly involved in the caudal fin regeneration of $tert$ mutant fish, we microinjected regenerating blastema of $tert^{-/-}$ fish with a construct that overexpresses the zebrafish $atrx$ gene under the control of the cytomegalovirus (CMV) constitutive promoter (Fig. 4E). Very strikingly, overexpression of $atrx$ gene impaired caudal fin regeneration compared to control-injected fish (Fig. 4F,G).

Altogether, these data demonstrate the involvement of ALT in the regenerative process in telomerase-deficient fish.

## Inhibition of nbs1 and atr genes affects caudal fin regeneration

In humans, the NBS1 protein, belonging to the MNR recombination protein complex (MRE11–RAD50–NBS1) implicated in the homolog recombination process, is required for both ALT phenotype development and extrachromosomal telomeric circles formation. Its inhibition decreased ALT-associated promyelocytic leukemia body numbers and decreased telomere length (Compton et al, 2007; Zhong et al, 2007). In the same way, ATR inhibition, which kinase activity is activated by NBS, disrupts ALT and selectively kill ALT cells in vitro (Flynn et al, 2015).

To further investigate whether telomere length was maintained through the ALT mechanism during the regeneration process, $nbs1$ and $atr$ were downregulated by injection of a vivo fluorescent morpholino (Mo), using a standard morpholino as negative control. Vivo morpholinos are oligomers used to knockdown a gene as the commonly used ATG or splice-blocking morpholinos, but modified for in vivo experiments. The knockdown efficiency of the $atr$ morpholino was characterized by Stern and colleagues (Stern et al, 2005). The injection of the $nbs1$ morpholino in zebrafish eggs resulted in the reduction of the expression of $nbs1$ mRNA at 3dpf (Fig. EV2A). Furthermore, routine PCR using cDNA as a template detected $nbs1$ mRNA species that retained the intron one of the gene as a result of the morpholino effect in blocking its splicing (Fig. EV2B). The injection was performed in the dorsal area of 48 hpa regenerated tissue of the caudal fin,

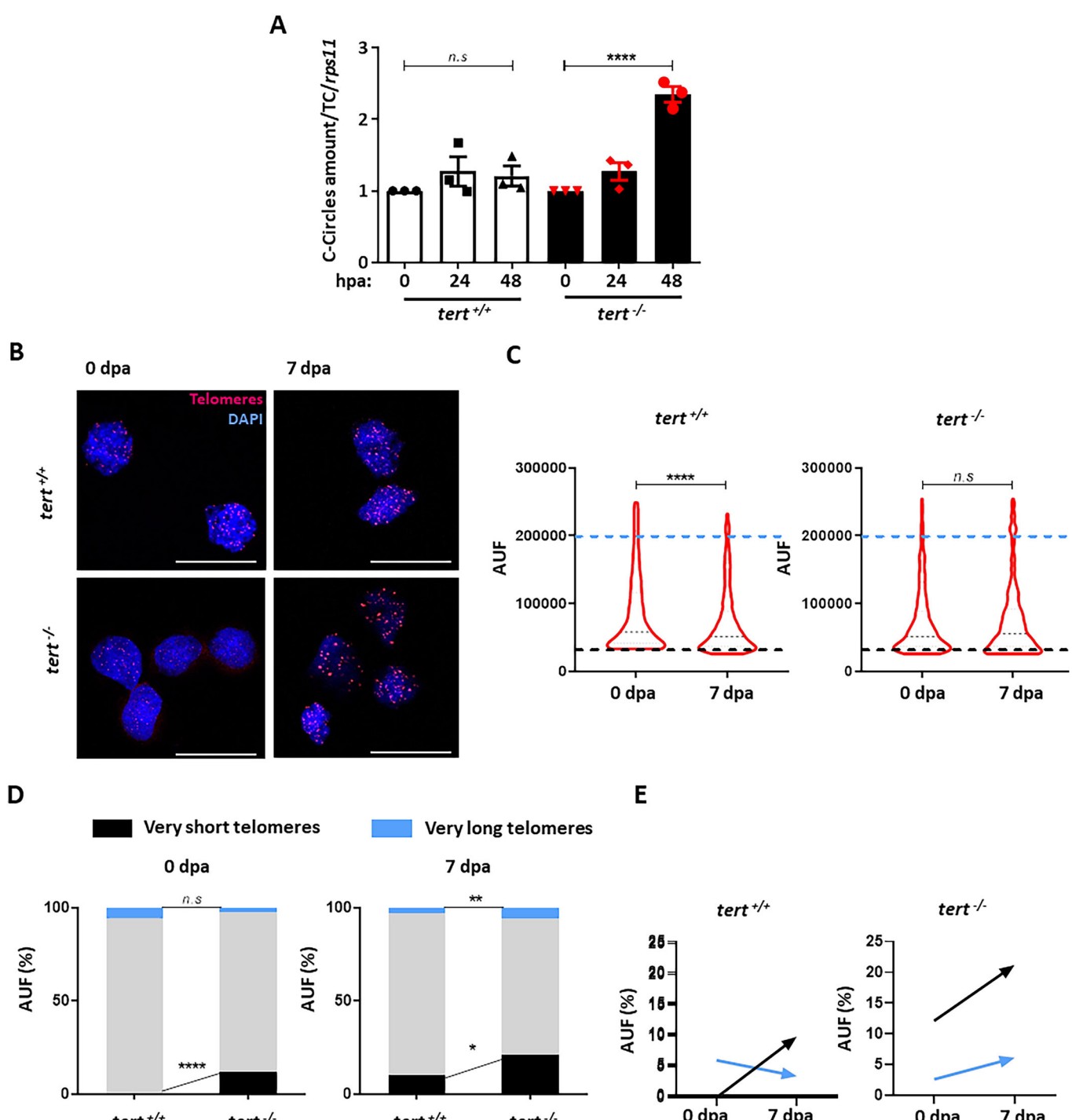

**Figure 3. Telomerase deficient zebrafish exhibit C-circles and heterogeneous telomere length in regenerative tissue.**

(A) C-circles abundance in regenerative tissue from wild type and telomerase-deficient adult zebrafish at 0, 24, and 48 hpa. $n = 3$ experiments. Each dot represents a biological replicate. (B) Representative images of Q-FISH in blastema cells from $tert^{+/+}$ and $tert^{-/-}$ genotypes at time 0 dpa and 7 dpa. Red staining shows telomeres and blue represents nuclei staining. Scale bar, 16,3 μm. (C) Telomeric fluorescence values (arbitrary units of fluorescence, AUF) of regenerative cells from $tert^{+/+}$ and $tert^{-/-}$ adult zebrafish at two times. $n = 1000$ cell events. (D) Telomeric fluorescence frequency of blastema cells from $tert^{+/+}$ and $tert^{-/-}$ adult zebrafish at 0 dpa and 7 dpa. Very long telomeres have a higher fluorescence of 200,000 AUF, and very short telomeres have a lower fluorescence of 30,000 AUF. (E) Schematic representation of the behaviour of very long (blue arrow) and very short telomeres (black arrow) during fin regeneration of wild type and $tert^{-/-}$ fish. All data were mean + s.e.m. n.s. not significant. ****$P < 0.0001$ for, one-way ANOVA plus Dunnett's post hoc test (A) and Student $t$-test (C, D). When exact $p$ values are not indicated is because GraphPad Prism software does not show it. Source data are available online for this figure.

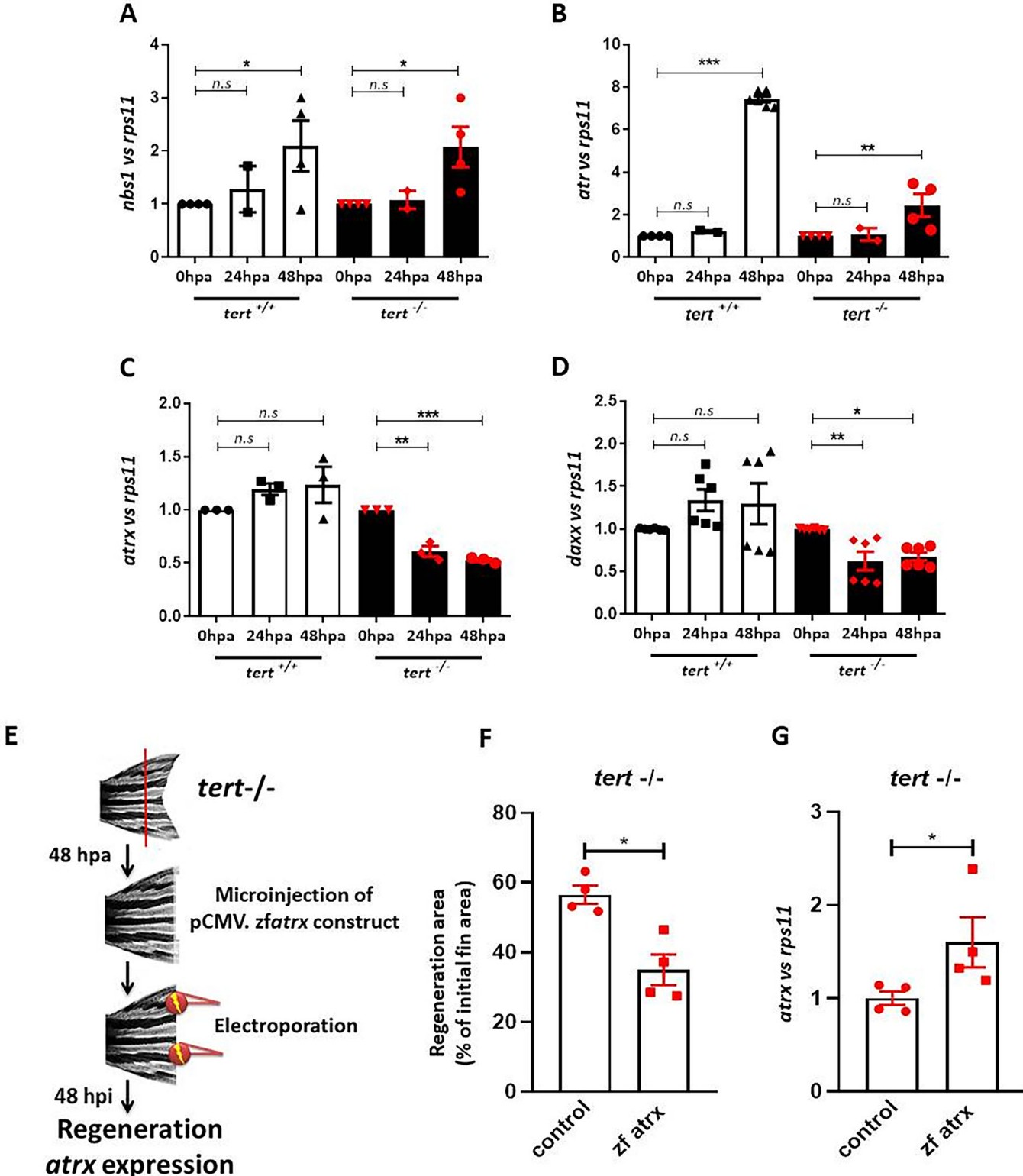

followed by electroporation of the injected zone to facilitate the morpholino entry into the cells. Forty-eight hours post injection (hpi), the regeneration rate was calculated, using the ventral area as an internal control (Fig. 5A).

Our results showed that inhibition of *nbs1* and *atr* genes decreases regeneration capacity in both genotypes, more remarkably in *tert^−/−^* compared to *tert^+/+^* zebrafish (Fig. 5B,C). Curiously, telomere length is maintained in the regenerated tissue in these fish

◄ **Figure 4. The expression of ALT-associated genes is modulated in regenerative tissue of wild type and telomerase-deficient zebrafish.**

(A) *nbs1* expression normalized to *rps11* expression in regenerative tissue from wildtype and telomerase-deficient adult zebrafish at 24 and 48 hpa. $n = 4$ experiments. Exact p values 0 hpa vs 48 hpa are $p = 0.04$ in $tert^{+/+}$ and $p = 0.044$ in $tert^{-/-}$. (B) *atr* expression normalized to *rps11* in regenerative tissue from $tert^{+/+}$ and $tert^{-/-}$ adult zebrafish at 24 and 48 hpa. $n = 4$ experiments. Exact p value 0 hpa vs 48 hpa in $tert^{-/-}$ is $p = 0.0019$. (C) *atrx* expression normalized to *rps11* in regenerative tissue from $tert^{+/+}$ and $tert^{-/-}$ adult zebrafish at 24 and 48 hpa. $n = 3$ experiments. Each dot represents a biological replicate. Exact p values in $tert^{-/-}$ are $p = 0.0097$ for 0 hpa vs 24 hpa are $p = 0.0025$ for 0 hpa vs 48 hpa. (D) *daxx* expression normalized to *rps11* in regenerative tissue from $tert^{+/+}$ and $tert^{-/-}$ adult zebrafish at 24 and 48 hpa. $n = 6$ experiments. Exact p values in $tert^{-/-}$ are $p = 0.0056$ for 0 hpa vs 24 hpa are $p = 0.0483$ for 0 hpa vs 48 hpa. (E) Schematic diagram of the workflow of the experiment. Regenerative fins of tert$-$/$-$ fish were microinjected 48 h post amputation (hpa) with a plasmid overexpressing the zebrafish *atrx* gene, and then electroporated. Forty-eight hours post injection (hpi), regenerated areas were quantified (F) and atrx gene expression was assessed (G). $n = 3$ experiments. Each dot represents a biological replicate. Exact p values are $p = 0.0286$ in (F) and $p = 0.0286$ in (G). All data were mean + s.e.m. n.s. not significant. *$P < 0.05$, **$P < 0.01$, and ***$P < 0.001$ for one-way ANOVA plus Dunnett's post hoc test in (A–D). *$P < 0.05$ for the Mann–Whitney test in (F, G). When exact p values are not indicated is because GraphPad Prism software does not show it. Source data are available online for this figure.

(Fig. EV3). To further show that the ALT mechanism is operating in the regeneration process in $tert^{-/-}$ and that is depending on *nbs1* and *atr* genes, we perform regeneration experiments in fish injected with the corresponding morpholinos and analysed the C-circle amount and telomeric repeat-containing (TERRA) RNAs levels (Fig. EV4A). TERRA RNAs are long non-coding RNAs transcribed from sub-telomeric regions of different chromosomes, containing telomeric sequences, and have been shown to be important for the preservation of telomere homeostasis and genomic stability (Kroupa et al, 2022). Furthermore, TERRA RNAs transcription has been also reported to destabilize telomere integrity, trigger break-induced replication at telomeres and promote telomere elongation in ALT cells in humans (Silva et al, 2021). As expected regeneration capacity of the caudal fins of tert$^{-/-}$ fish microinjected with both morpholinos decreased compared to control fish injected with the Mo-std (Fig. EV4B–D). Consistently, TERRA RNAs levels, as well as C-circles amount, were increased in the regenerating tissue, and this induction was lower when the expression of *atr* and *nbs1* was decreased by Mo-injection (Fig. EV4E,F).

To understand the contribution of ALT and telomerase to the regenerative process, a regeneration experiment similar to the previous one was carried out in wild-type zebrafish. *atr* (to inhibit ALT) and *tert* (to inhibit telomerase) vivo fluorescent morpholinos were microinjected into the dorsal area of 48 hpa regenerated tissue (Fig. 5D,E). The *tert* morpholino knockdown efficiency has been already showed (Imamura et al, 2008). Regeneration capacity was impaired also in these experimental conditions. To further show that ALT could be active in these conditions, we performed caudal fin amputation in wild-type fish and microinjected the blastema with the indicated mix of morpholinos and analyzed the levels of TERRA RNAs transcription. The regeneration capacity is inhibited in tissues injected with a mix of Mo-std+Mo-tert, a mix of Mo-std+Mo-atr, or a mix of Mo-tert + Mo-atr compared with a control injected with a double dosis of Mo-std (std 2x, Fig. EV4G). In addition, the expression of *tert* and *atr* is decreased in the regenerated blastema upon morpholino injection (Fig. EV4H,I), indicating that the genetic inhibition of the expression of these genes was efficient. Finally, the levels of TERRA RNAs are increased upon amputation, and this induction is reduced when we Mo-*atr* or a combination of Mo-*atr* + Mo-*tert* were microinjected (Fig. EV4J). Taking altogether, these results would indicate that the ALT mechanism is induced upon an injury and operating in the regenerating tissue of both wild-type and *tert*-deficient fish. Therefore, telomerase and ALT mechanism would be

involved in zebrafish caudal fin regeneration, and when fish do not have one of these mechanisms, they could use the other to regenerate.

## Pharmacological inhibition of ATR in zebrafish adults and larvae affects regeneration

We tested the effect of pharmacological inhibition of ALT in this model using ATR Inhibitor IV, an ATR-selective inhibitor (Flynn et al, 2015). The effect of ATR inhibition during the caudal fin regeneration process was analysed by a regeneration assay on wild-type and *tert*-deficient adult zebrafish, treating fish with 10 and 50 μM of the inhibitor for 24 and 48 h (Fig. 6A). Treatment with ATR inhibitor IV decreased the regenerative capacity of wild-type and *tert*-deficient fish (Fig. 6D,E). The efficiency of ATR inhibition in these experiments was analysed by western blot against ATR/ATM phosphorylated substrates (Fig. 6B,C).

Since zebrafish larvae are a powerful tool for high-throughput screening, we developed a regeneration assay on larvae for pharmacological screening. We treated 48 h post fertilization (hpf) zebrafish larvae with 10 μM ATR inhibitor IV by immersion immediately after tail amputation (Fig. EV5A). Twenty-four hours later (72 hpf larvae), the regeneration area was measured. As expected, the larvae treated with the ATR inhibitor showed a smaller regenerated area compared to untreated larvae, although only statistically significant for telomerase-deficient genotypes (Fig. EV5B,C).

Therefore, *atr* inhibition decreased the regenerative capacity of the zebrafish's caudal fin, more significantly in telomerase-deficient fish. This means that the caudal fin regeneration assay in $tert^{-/-}$ zebrafish could be used as a fast and simple model to identify drugs that inhibit ALT activity.

## Discussion

Regeneration is a complex process of cell renewal, restoration and tissue growth. High proliferation rates are accompanied by telomerase activity to maintain telomere length to sustain replication and cell division.

Zebrafish has become a powerful model for investigating regenerative processes because of its capacity to completely repair several organs after injury.

An amputated caudal fin is covered within the first several hours by epidermis, and within 1 to 2 days, a regeneration blastema

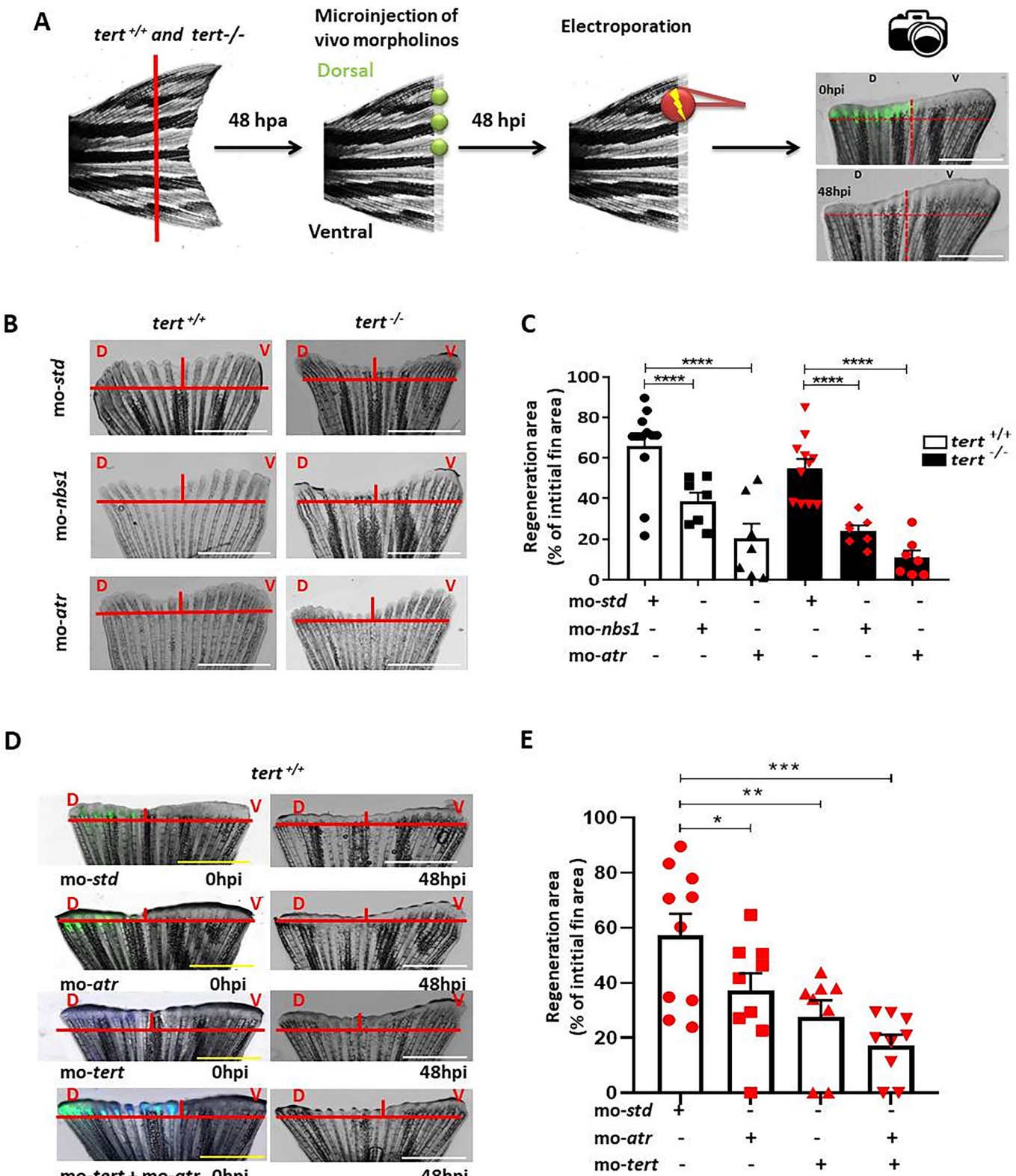

forms. The blastema is a proliferative mass of morphologically similar cells, formed through disorganization and distal migration of fibroblasts and osteoblasts (or scleroblasts) proximal to the amputation plane (Gemberling et al, 2013). High proliferation rates accompanied by telomerase attrition require telomere maintenance to sustain replication and cell division. However, the involvement of telomerase and telomeric length behaviour has not been characterized during caudal fin regeneration.

**Figure 5. Inhibition of *nbs1* and *atr* affects adult caudal fin regeneration.**

(A) Schematic representation of vivo fluorescent morpholino microinjection into the dorsal area of the caudal fin after 48 hpa. The microinjected area is then electroporated. The ventral area is used as a control of normal regeneration. After 48 h post injection (hpi), the regeneration area was calculated. Scale bar, 1 mm. (B) Representative pictures of caudal fin regeneration after 48 h post-morpholino microinjection. Scale bar, 4.6 mm. (C) Regeneration area of caudal fin in $tert^{+/+}$ and $tert^{-/-}$ genotypes with *nbs1* and *atr* knockdown. $n = 3$ experiments. Each dot represents a biological replicate. (D) Representative images of caudal fin at 0 hpi of vivo morpholinos (standard and *atr* in green and *tert* in blue) in the dorsal half (D), and after 48 hpi. (V) uninjected ventral half. Scale bar, 1 mm. (E) Regeneration area of caudal fin in wild-type fish with *atr* and/or *tert* knockdown. $n = 3$ experiments. Each dot represents a biological replicate. Exact $p$ values are $p = 0.0291$ for control vs mo-atr, $p = 0.0053$ for control vs mo-tert and $p = 0.0002$ for control vs mo-atr + mo-tert. All data were mean + s.e.m. *$P < 0.05$, **$P < 0.01$, ***$P < 0.001$, and ****$P < 0.0001$ for one-way ANOVA plus Sidak´s post hoc test in (C) and plus Holm–Sidak´s post hoc test in (E). When exact $p$ values are not indicated is because GraphPad Prism software does not show it. Source data are available online for this figure.

Our results confirmed the notable regeneration capacity of the zebrafish after a single or successive caudal fin amputations (Anchelin et al, 2011; Azevedo et al, 2011; Shao et al, 2011), even in telomerase-deficient zebrafish. Curiously, the regeneration area reaches more than 100% of the initial caudal fin size at 20 dpa in all fish phenotypes. This could be due to the continuous growth of the fish, particularly when the fish were incubated at the experimental temperature of 32 °C.

Surprisingly, telomerase-deficient zebrafish maintained its regenerative ability across time in both 4-month-old young fish and in prematurely aged 11-month fish. Although the regenerative capacity was not lost in old fish, it was reduced both after a single clip and by successive clips. Unexpectedly, they were able to keep the length of their telomeres constant after successive clips. It could be argued that a few divisions in stem cells may be sufficient to ensure the regeneration of the fin, thus not affecting the total telomere length even in the absence of TERT. However, the experiments involving 12 consecutive cycles of fin amputation and regeneration (7 days each cycle) show that $tert^{-/-}$ fish are capable of regenerating caudal fin at a similar level as wild-type fish do. This represents 84 days ($12 \times 7$) of continuous cell divisions, and would result in telomere length attrition and impaired regeneration capacity, suggesting that although telomerase has an important role in regeneration, the participation of an alternative mechanism of telomere lengthening in zebrafish telomerase mutants is essential.

In the case of telomerase deficiency, homologous recombination constitutes an alternative method (ALT "alternative lengthening of telomeres") to maintain telomere DNA. Rolling-circle replication may use extrachromosomal t-circles, as has been shown in human ALT cells and in a wide variety of organisms, including yeasts, higher plants and *Xenopus laevis*, to elongate telomere. From an evolutionary perspective, this widespread occurrence of t-circles may not only represent a backup in the event of telomere dysfunction, but may be the primordial systems of telomere maintenance (Fajkus et al, 2005).

The presence of extrachromosomal telomeric circles, proposed to result from homologous recombination events at the telomeres, has been used as a marker of ALT cells. At 24 hpa, we detected telomeric DNA circles in the regenerative cells from the wild-type and the mutant zebrafish $tert^{-/-}$, although the amount of telomeric DNA circles is significantly higher in the mutant fish at 48 hpa, pointing out that ALT has been activated in these cells.

Moreover, other characteristics of ALT cells include a highly heterogeneous chromosomal telomere length (Murnane et al, 1994), which was observed after 7 dpa in $tert^{-/-}$, where the percentage of much shorter telomeres and very long telomeres were increased two times. However, in control fish, the percentage

of very short telomeres increased and the percentage of very long telomeres decreased, suggesting a different telomere maintenance dynamics. ALT activation (i.e., C-circles increase) is expected to happen, and in fact detected, very early in the regeneration process (Gao and Pickett, 2022) and eventually results in telomere length heterogeneity several days after amputation, when a lot of rounds of cell divisions have occurred.

Although the mechanism and causes of ALT are still not well-known, we wanted to study the expression of proteins involved in ALT during the regeneration process, including proteins involved in DNA repair as NBS1, a critical regulator of recombination recruited by RPA, ATR complex, and chromatin remodelling complexes DAXX, and ATRX, which mutations are associated with ALT (Hu et al, 2016; Lovejoy et al, 2012; Napier et al, 2015). Curiously, we observed an increased gene expression of ALT activator proteins and a decrease in ALT inhibitor proteins in both wild-type and telomerase-deficient zebrafish, suggesting that the main players of ALT and their mechanisms are conserved during evolution, and that both mechanisms of telomere maintenance could coexist in the regeneration process in wild-type fish. *atrxx* and *daxx* expression is downregulated in regenerating tissue of the *tert* mutant fish, whereas in wild-type animals, does not change. Very interestingly, overexpression of *atrx* in regenerating blastema decreased regeneration capacity of *tert* mutant fish, indicating that the ALT mechanism responsible for caudal fin regeneration in $tert^{-/-}$ fish is facilitated, at least in part, by low *atrx* levels. The mechanisms behind the downregulation of *atrx* and *daxx* require further research. This could be due to the fact that TERT is supporting the expression of these genes upon fin amputation, preserving the telomere integrity and assuring that telomere conservation during the regeneration process in these animals is mainly mediated by telomerase complex activity. This could be achieved by the well-known role of TERT as a transcription factor. Furthermore, the absence of telomerase activity may also promote the downregulation of the expression of *atrx* and *daxx*.

Moreover, by inhibiting *nbs1* or *atr* by *vivo* morpholinos or ATR pharmacologically, thus preventing the maintenance of telomere length by ALT, a lower regeneration efficiency was observed during caudal fin regeneration in wildtype and telomerase-deficient zebrafish, although the effect is always more evident in $tert^{-/-}$ genotype. In addition, the increase in TERRA RNA levels, which has been shown to trigger the ALT mechanism in human cells, decreases to T0 levels upon *nbs1* and *atr* inhibition in *tert*-deficient animals. We also observed a decrease in the C-circles amount in *nbs1* and *atr*-morpholino injected fish, although not decreasing to base levels. This could be due to the inherent differences in the stability of these molecules. It is well-established that circular

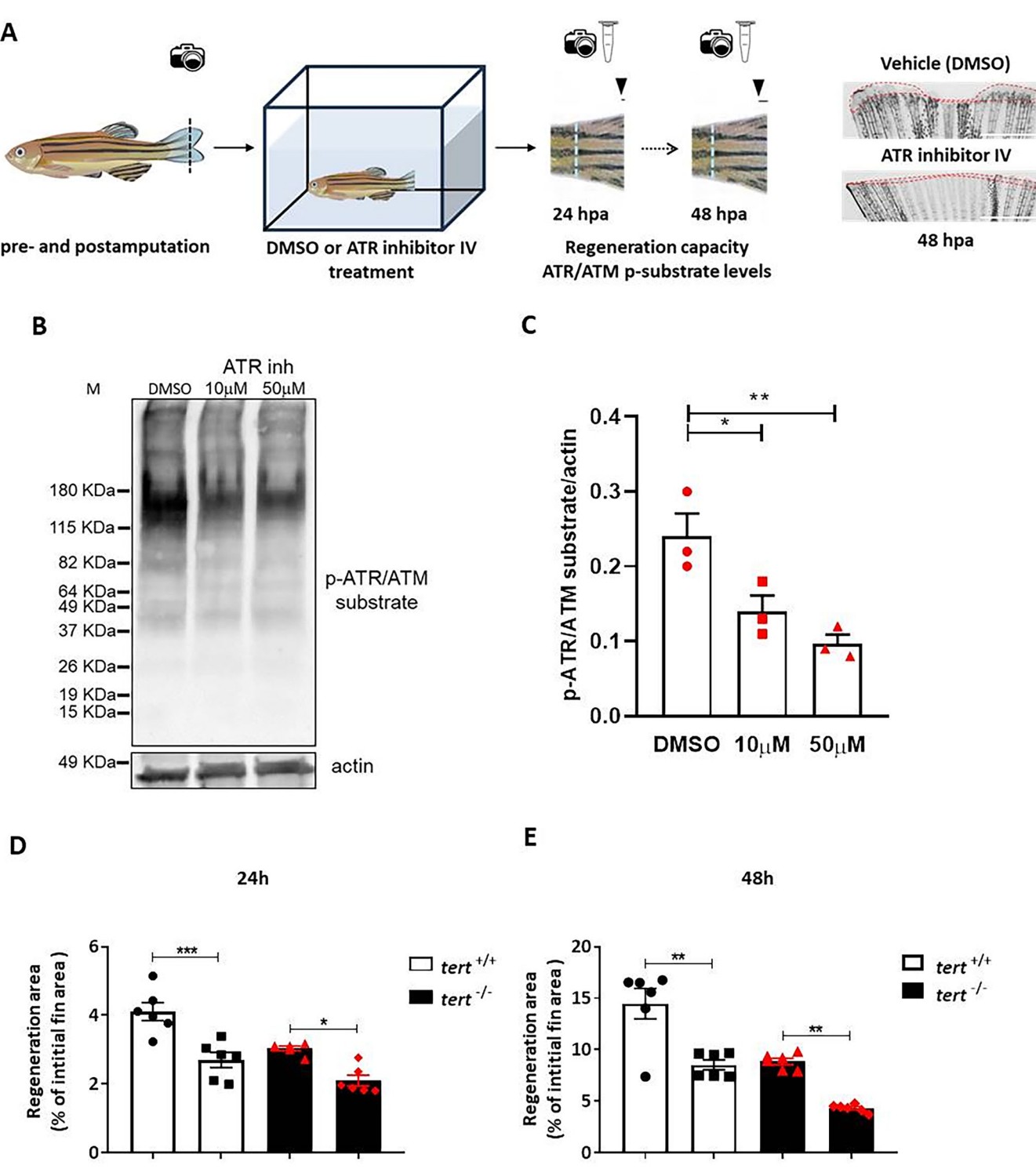

**A** pre- and postamputation — DMSO or ATR inhibitor IV treatment — 24 hpa → 48 hpa — Regeneration capacity ATR/ATM p-substrate levels — Vehicle (DMSO) / ATR inhibitor IV / 48 hpa

**B** p-ATR/ATM substrate / actin

**C** p-ATR/ATM substrate/actin

**D** 24h

**E** 48h

extrachromosomal DNA, such as C-circles, is structurally more stable than linear DNA and even more so than RNA. In contrast, TERRA RNA is subject to more dynamic regulation and degradation, likely mediated by various regulatory proteins. Therefore, it can be speculated that while C-circles remain relatively stable due to their structural properties, TERRA RNA is under stricter control, leading to its faster normalization during tissue regeneration.

*tert* and *atr* knockdown showed an impaired regeneration percentage, as well as a decrease in the expression of TERRA RNAs in wild-type fish. We observed a slight difference in the inhibition of the regeneration area in wild-type animals injected with Mo-

**Figure 6. Pharmacological inhibition of ATR in adults affects regeneration.**

(A) Experimental design of regeneration assay in adult zebrafish. ATR inhibitor IV is administered to adult fish tanks immediately after tail fin amputation. Regenerative capacity (arrowhead and line) and ATR/ATM p-substrate levels are tested at 24 and 48 h post amputation (hpa), compared with controls treated with DMSO. The caudal peduncle is indicated in a blue dotted line. On the right, representative pictures of caudal fin regeneration after 48 h post-treatment. Scale bar, 1 mm. (B) Representative WB for detection of phosphorylated ATR/ATM substrates in regenerating blastema of *tert* mutant adult fish. (C) Quantification of the WB. The intensity of the whole lane was used. n = 3 experiments. Exact p values are p = 0.0348 for DMSO vs 10 μM and p = 0.0072 for DMSO vs 50 μM. (D) Regeneration area of the caudal fin in adults at 24 hpa. n = 3. Each dot represents a biological replicate. Exact p value for control vs ATR inhibitor is p = 0.0053 in tert$^{-/-}$. (E) Regeneration area of the caudal fin in adults at 48 hpa. n = 3. Each dot represents a biological replicate. Exact p value for control vs ATR inhibitor is p = 0.0013 in tert$^{-/-}$. All data were mean + s.e.m. *P < 0.05, **P < 0.01, and *** P < 0.001 for one-way ANOVA plus Sidak´s post hoc test. When exact p values are not indicated is because GraphPad Prism software does not show it. Source data are available online for this figure.

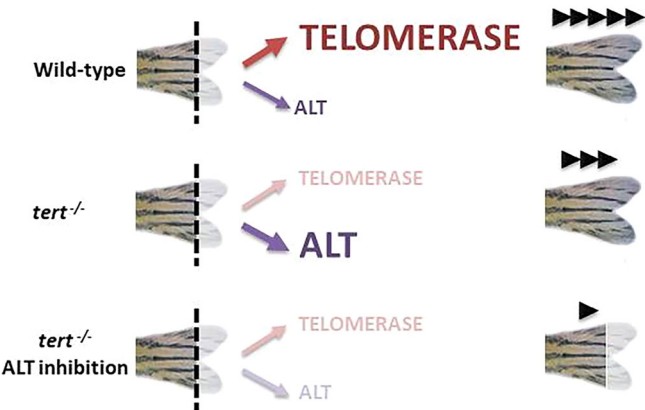

**Figure 7. Schematic representation of telomeric maintenance in the zebrafish caudal fin.**

Telomerase is the main telomeric maintenance mechanism that allows the regeneration of the tail fin after damage. However, in telomerase-deficient fish, regeneration occurs thanks to ALT, although more slowly. Therefore, if both mechanisms are inhibited, no regeneration of the zebrafish caudal fin is possible.

*tert* + Mo-*atr* compared to *tert* mutants injected with Mo-*atr*. This difference could be due to a partial efficacy of inhibition of the *tert* expression when the Mo was used.

We observed that telomere length was unaffected in all the morpholino injection conditions. This may be attributed to the activation of alternative telomere maintenance mechanisms that remain unaffected by the morpholinos used. Specifically, while the morpholino targeting the nbs1 and atr genes effectively disrupts the canonical ALT pathway, the morpholino targeting tert inhibits telomerase-mediated telomere lengthening. Consequently, the inhibition of one mechanism may compensate for the other, allowing for the maintenance of telomere length during the regeneration process. This complexity highlights the interplay of multiple pathways involved in sustaining telomere length under these experimental conditions.

The regenerating cells in *tert*$^{-/-}$ present all the characteristics of ALT-positive cells, such as telomeric heterogeneity, presence of extrachromosomal telomeric circles and overexpression of genes involved in homologous recombination.

Altogether, our results lead us to conclude that in the absence of telomerase, telomere length is maintained through a homologous recombination mechanism during the early phases of regeneration. By contrast, in the wild-type zebrafish, the two mechanisms

(telomerase-dependent and telomerase-independent) could coexist and participate in the maintenance of telomeres, with ALT inhibition causing less severe effects than telomerase inhibition. During zebrafish aging, dysfunctional telomeres could lead to the emergence of telomere recombination and so to ALT activation without hampering telomerase function (Brault and Autexier, 2011). The inhibition of both mechanisms greatly decreases the regeneration ability of the caudal fin (Fig. 7). Coexistence of these two mechanisms has been described in cancer ex vivo and in vitro (De Vitis et al, 2018). The capability of some cancer cells ex vivo to "switch" from one mechanism to the other is important, in a particular way, for the application of dual targeted (anti-telomerase and anti-ALT) therapy to avoid drug resistance and will promote better therapies. Importantly, zebrafish tail regeneration provides the first in vivo model to study both telomere maintenance mechanism and their coexistence.

Therefore, the ALT mechanism allows the regeneration of the caudal fin in telomerase-deficient zebrafish. However, not all zebrafish tissues can regenerate in the absence of telomerase, for example telomerase is essential for zebrafish heart regeneration (Aix et al, 2018; Bednarek et al, 2015), where that absence of telomerase impairs proliferation leading to the accumulation of DNA damage, and *tert*$^{-/-}$ zebrafish hearts acquire a senescent phenotype (Bednarek et al, 2015). This suggests that telomere maintenance mechanisms may be specific to tissues and organs. In addition, the coexistence of both telomere maintenance mechanisms is also specific to the biological process studied, since a zebrafish model of cancer that uses ALT for telomere maintenance, reverted all ALT features when *tert* was over-expressed (Idilli et al, 2020a), as has been described before in humans (Perrem et al, 1999). Further studies are needed to elucidate a possible evolutionary explanation for these differences. One hypothesis would be that fins as external organs are crucial for the survival of the species and highly exposed to external damage and predators, so the existence of both mechanisms ensures telomere maintenance in cells implicated in the regeneration after injury of an organ essential for fish survival.

ALT is only active in about 10% of tumors, but these are tumors with a poor prognosis. ALT mechanism is a potential target for therapy (Zhang and Zou, 2020). A zebrafish model of cancer that uses ALT for telomere maintenance was recently described (Idilli et al, 2020a; Idilli et al, 2020b). However, an ALT model of regeneration has never been described and would be a powerful tool to test drugs. Zebrafish has proven to be an indisputably useful vertebrate model system, with great advantages over other models, such as the ease, cost and speed of screening compounds (Murphey and Zon, 2006; Yoganantharjah and Gibert, 2017). In this work, the zebrafish larvae has been shown to be a promising model for testing drugs that inhibit ALT and/ or telomerase, paving the way to be used in large-scale screening.

On the other hand, we confirmed the impressive capability of the zebrafish to heal and regenerate fin tissues after an injury, only limited by aging, even in the absence of telomerase, and the involvement of ALT mechanisms in the telomere length maintenance during the regeneration process of the zebrafish caudal fin. To our knowledge, this is the first tissue of an animal model demonstrating the coexistence of telomerase and ALT mechanism, only found in some cancer populations and engineered cellular models (De Vitis et al, 2018). Zebrafish has proven to be an exceptionally useful model for further studies on the role of telomeres and telomerase in regeneration.

# Methods

### Reagents and tools table

| Reagent/resource | Reference or source | Identifier or catalog number |
|---|---|---|
| **Experimental models** | | |
| Zebrafish | Zebrafish International Resource Center (ZIRC) and Sanger Institute | |
| HeLa | Gift Maria A. Blasco | RRID: CVCL_0030 |
| SaOS-2 | ATCC | RRID: CVCL_0548 |
| **Recombinant DNA** | | |
| pCMV.zfATRX | GeneScript | |
| **Antibodies** | | |
| Rabbit phospho-(Ser/Thr) ATM/ATR substrate | Cell Signaling | 2851 |
| Sheep anti-rabbit IgG- HRP | Cytiva | NA934-1ML |
| **Oligonucleotides and other sequence-based reagents** | | |
| **PCR primers (gene/seq.)** | Merck | |
| *Telom* F: 5′-ggttttttgagggtgagggtgagggtgagggtgagggt-3′ R: 5′-tcccgactatccctatccctatccctatccctatcccta-3′ | Lau et al, 2013 Wang et al, 2017 | |
| *rps11* F:5′-acagaaatgccccttcactg-3′ R:5′-gcctcttctcaaggttg-3′ | García-Castillo et al, 2021 | |
| *tert* F: 5′-cggtatgacggccyatcact-3′ R: 5′-taaacggcctccacagagtt-3′ | Anchelin et al, 2013 | |
| *nbs1* F:5′-ttcagagcatgcagttggac-3′ R: 5′-tccggaattggaggttga-3′ | Merck KiCq Start® Primers | |
| *daxx* F: 5′-aagatctgacctgtctgaag-3′ R: 5′-gtgttgctacgtttacctta-3′ | Merck KiCq Start® Primers | |
| *atrx* F: 5′-gacgataaagtgttggtgtt-3′ R: 5′-gtcttcaatgaggtcaagag-3′ | Merck KiCq Start® Primers | |
| *atr* F: 5′-tgaacaccaggtataagagc-3′ R: 5′-tcaactttgaccaattcacc-3′ | Merck KiCq Start® Primers | |
| **Genotyping primers (name/seq.)** | Merck | |
| **tertFwt** 5′-tgccggaggtcttggccg-3′ | | |
| **tertRwt** 5′-cgcacacctgcagaaca-3′ | | |
| **tertRmut** 5′-cgcacacctgcagaact-3 | | |
| **Morpholinos (gene/seq.)** | GeneTools | |
| *nbs1* 5′-gatttacacagagaagattacctcc-3′ | | |
| *atr* 5′-tgacattttctagtccttgctccatc-3′ | Stern et al, 2005 | |
| *tert* 5′-ctgtcgagtactgtccagacatctg-3′ | Imamura et al, 2008 | |
| **Negative control** 5′-cctcttacctcagttacaatttata-3′ | | |
| **Chemicals, enzymes and other reagents** | | |
| Phusion high-fidelity DNA polymerase | ThermoFisher | F530L |
| phi29 DNA polymerase | New England Biolabs | M0269L |
| SuperScript IV VILO Master Mix | Thermo Fisher | 11755050 |
| 2x SYBR Premix Ex Taq Mix | TAKARA | RR420A |
| RPMI 1640 | Merck | R0883 |
| DMEM high glucose | BioWest | L-0101 |
| Fetal bovine serum (FBS) | Cytiva | SV30160.03 |
| Cellstar cell culture six-well plate | Grenier bio-one | 657-160 |
| Benzocaine | Merck | E1501 |
| Colagenase type I | ThermoFisher | 17100017 |
| 100-μm cell strainer mesh | Falcon | 352360 |
| 40-μm cell strainer mesh | Falcon | 352340 |
| Bovine serum albumin (BSA) | New England Biolabs | B9200S |
| FITC-labeled telomeric PNA probe | PNA Bio | F1009 |
| ATR inhibitor IV | Calbiochem | 5049720001 |
| DMSO | Merck | D8418 |
| Tween-tris-buffered saline 10x | DAKO | S3006 |
| MycoProbe Mycoplasma Detection kit | R&D Systems | CUL001B |
| AmpFLSTR™ Identifiler Plus PCR Amplification Kit | Thermo Fisher | 4427368 |
| ChargeSwitch gDNA Micro Tissue Kit | Thermo Fisher | CS11203 |
| PureLink RNA Mini kit | Thermo Fisher | 12183025 |
| Nitrocellulose membrane, 0.45 μm | Bio-Rad | 1620117 |
| **Software** | | |
| GraphPad Prism 7.0 | https://graphpad.com | |
| LasX | Leica | |
| ImageJ | https://imagej.net/ij/ | |
| **Other** | | |

| Reagent/resource | Reference or source | Identifier or catalog number |
|---|---|---|
| Leica M205 FA fluorescence stereo microscope and DFC365FX camera | Leica | |
| NEPAGENE electroporator | | |

## Ethics statement

The experiments performed comply with the Guidelines of the European Union Council (Directive 2010/63/EU) and the Spanish RD 53/2013. Experiments and procedures were performed as approved by the Bioethical Committee of the University Hospital "Virgen de la Arrixaca" (210621/4/ISCIII).

## Animals

Zebrafish (*Danio rerio* H., Cypriniformes, Cyprinidae) were obtained from the Zebrafish International Resource Center (ZIRC) and mated, staged, raised and processed using standard procedures. Details of husbandry and environmental conditions are available on protocols.io (Lawrence et al, 2018). *tert* mutant line (allele hu3430) was obtained from the Sanger Institute.

The animal sample size was selected according to the Refinement rule when working with experimental animals. The experiments were performed blinded.

## *tert* mutant genotyping

For genotyping *tert* mutants, fish were anesthetized with 0.05% benzocaine (Sigma), and tissue was obtained by cutting the apical part of the tail caudal fin. The tissue was incubated in alkaline lysis solution (NaOH 25 mM, EDTA 0,2 mM, pH 12) at 95 °C for 20 min, chilled on ice, and neutralizing solution (Tris-HCl 40 mM, pH 5) was added. After centrifugation, the supernatant was used as a template for a PCR using Phusion high-fidelity DNA polymerase (Thermo Fisher), and tertFw forward primer and the tertRwt (for wt allele detection) or tertRmut (for mutant allele detection) reverse primers. Amplification protocol consisted of a denaturing step of 4 min at 94 °C, 40 cycles of 94 °C 1 min, annealing at 65 °C for 30 s, and extension at 72 °C for 30 s more, and a final step of 72 °C for 10 min. The sequence of the primers is listed in the Reagents and tools table.

## Caudal fin regeneration assay on adult zebrafish

For the caudal fin regeneration assay, zebrafish from $tert^{+/+}$, $tert^{+/-}$, and $tert^{-/-}$ genotypes were anesthetized with 0.05% benzocaine (Sigma). Fin tissue was removed ~2 cm from the base of the caudal peduncle, proximal to the first lepidotrichial branching point, using a razor blade. For the single amputation assay or the successive amputations assay, images of zebrafish caudal fins were taken before and after the amputation, as well as from different days post amputation (dpa) following the respective experimental design. The fin tissue was preserved and processed accordingly in each experiment. When the fish recovered, they were returned to recirculating water heated to 32 °C for the duration of the

experiment, to accelerate the regeneration process and assure that the amount of regenerated blastema is enough to perform the appropriated analysis. Each fish was tracked and imaged individually to calculate regeneration progress over time. The percent of caudal fin regeneration was determined based on the caudal fin area after regeneration divided by the original area of the non-amputated caudal fin. We used a total of 6–12 fish for each genotype group, divided into at least two independent experiments.

## Cell isolation

Regenerated portion of caudal fin tissue obtained from zebrafish at different clips were incubated in Trypsin (0.5 mg/mL)/EDTA (0.1 mg/mL, Gibco) in PBS for 1 min (min), centrifuged (600×*g*, 5 min) and then incubated in colagenase (0.5 mg/mL) (Gibco) in RPMI medium (Lonza) supplemented with $CaCl_2$ $2H_2O$ (0.7 mg/mL) for 30 min. The cell suspensions were obtained by pipetting, smashing and filtering the digested tissues through a 100-µm mesh and a 40-µm mesh (Falcon) successively, finally washed and resuspended in PBS.

## Flow-FISH

Cells from each caudal fin sample obtained as described before were washed in 2 mL PBS supplemented with 0.1% bovine serum albumin (BSA, NEB). Each sample was divided into two replicate tubes: one pellet was resuspended in 500 ml hybridization buffer, and another in hybridization buffer without FITC-labeled telomeric PNA probe (PNABio) as a negative control. Samples were then denatured for 10 min at 80 °C under continuous shaking and hybridized for 2 h in the dark at room temperature. After that, the cells were washed twice in a washing solution [70% deionized Formamide (Sigma), 10 mM Tris (Sigma), 0.1% BSA (NEB) and 0.1% Tween-20 (Sigma) in $dH_2O$, pH 7.2]. The cells were then centrifuged, resuspended in 500 mL of propidium iodide solution, incubated for 2 h at room temperature, stored at 4 °C and analysed by flow cytometry within the following 48 h.

## Q-FISH

Cell suspensions from regenerative tissue were obtained as described before and fixed in methanol:acetic acid (3:1). Inter-phases were spread on glass slides and dried overnight, and FISH was performed as described in Canela et al (Canela et al, 2007). Cy3 and DAPI images were captured at 100x magnification using a confocal microscope (LSM 510 META from ZEISS, Jena, Germany) and Zeiss Efficient Navigation (ZEN) interface software. Telomere fluorescence signals were quantified using ImageJ software.

## Cell culture

The cell lines HeLa 293-C and HeLa 1121-L were derivatives of the human HeLa cell line (RRID: CVCL_0030), and were a gift from Dra. Maria A. Blasco from CNIO, Spain. SaOS-2 human cell line (RRID: CVCL_0548) was obtained from ATCC (cat. HTB-85).

Cells were grown in Dulbecco's modified eagle medium (DMEM, Biowest) supplemented with 10% fetal bovine serum (FBS, Cytiva), 1% Penicillin/Streptomycin mix (Lonza) at 37 °C in a humidified incubator with 5% $CO_2$. Mycoplasma-free conditions

were tested every 3 months using the MycoProbe Mycoplasma Detection kit (R&D Systems), and all experiments were performed with mycoplasma-free cells. Cell lines have been authenticated within the last three years in the Cell and Tissue Culture facility of the University of Murcia (Molecular Biology Service) using the AmpFLSTR™ Identifiler™ Plus PCR Amplification Kit (Thermo Fisher Scientific).

## C-Circles assay

The assay for the detection of telomeric DNA circles was performed as described (Lau et al, 2013) with modifications. Caudal fin tissues of adult animals were collected from $tert^{+/+}$ and $tert^{-/-}$ genotypes at 0 h post amputation (hpa), and from the corresponding regenerate portion at 24 and 48 hpa were preserved at −80 °C. Genomic DNA (gDNA) was extracted from the caudal fin samples using the ChargeSwitch™ gDNA Micro Tissue Kit (Invitrogen). A rolling-circle amplification reaction (RCA) of partially double-stranded C-circles was performed with 0.2 μg/μL bovine serum albumin (NEB), 0.1% Tween-20 (Sigma), 4 μM dithiotreitol (DTT) (Invitrogen), 1 μM dNTPs (Invitrogen), 3.75U φ29 polymerase (NEB), 1X φ29 polymerase buffer (NEB), and 64 ng of gDNA. The mix was incubated at 30 °C for 8 h and then at 65 °C for 20 min. Finally, the RCA reaction was diluted to 40 μL with water. For each sample, the assay was done with and without φ29 polymerase, capable of amplifying circular DNA in a self-primed way.

After that, the telomeric sequences were detected through real-time PCR using the C-circles amplification reaction as template. Real-time PCR was performed with a StepOnePlus instrument (Applied Biosystems) using SYBR® Premix Ex Taq™ (Perfect Real Time) (Takara). The primers (Telom F-Telom R) used are in the Reagents and Tools Table. The reaction mix and qPCR protocol used were described in (Lau et al, 2013). Finally, a second real-time PCR was performed with the same RCA to detect ribosomal protein s11 (*rps11*) as a single-copy gene (SCG) used to normalize. The primers used are in the Reagents and Tools Table. Both qPCRs were performed using the RCA with and without φ29 polymerase amplification as template.

As a reference assay, the same protocol was performed using gDNA obtained from ALT-positive human osteosarcoma tumor cell line (Saos-2) and ALT-negative human cervical tumor cell line (HeLa). In this case, the single-copy gene reference was the gene *36B4*. The primers used for *36B4* amplification and the qPCR conditions are described in (Lau et al, 2013).

The C-circles amount value was obtained as follows: each telomeric qPCR was normalized to the corresponding single-copy gene qPCR. Then, the normalized value of the samples without φ29 polymerase (telomere content per single-copy gene) was subtracted from that with φ29 polymerase, and this final value was named as NORMTEL. The C-circles amount per telomere content per single-copy gene (CC amount/telomere content/SCG) value is $2^{NORMTEL}$ (Lau et al, 2013).

## Telomere length measured by qPCR

Relative telomere length was measured by qPCR using telomF-R primers (Reagents and Tools Table) and 16 ng of genomic DNA as template. The qPCR protocol is described elsewhere (Lau et al, 2013).

## Gene expression analysis

Total RNA was extracted from the homogenized regenerative tissue in TRIzol Reagent (Ambion) and using the PureLink RNA Mini kit (Ambion), following the manufacturer's instructions. RNA was treated with DNase I, amplification grade (1 U/mg RNA; Thermo Fisher Scientific). cDNA was generated by the SuperScript™ IV VILO™ Master Mix (Invitrogen), following the manufacturer's instructions. Real-time PCR was performed with a StepOnePlus instrument (Applied Biosystems) using SYBR® Premix Ex Taq™ (Perfect Real Time) (Takara). Reaction mixtures were incubated for 30 s at 95 °C, followed by 40 cycles of 5 s at 95 °C, 20 s at 60 °C, and finally a melting curve protocol. We use the qPCR protocol described by Wang et al, 2017 for the detection of TERRA RNAs (Wang et al, 2017). *rps11* content in each sample was used for normalization of zebrafish mRNA expression, using the comparative *Ct* method ($2^{-\Delta Ct}$). The primers used are shown in the Reagents and Tools Table.

## Morpholino and plasmid injection and electroporation

Adult $tert^{+/+}$ and $tert^{-/-}$ zebrafish were anesthetized in 0.05% benzocaine prior to the amputation of their caudal fins. Following the surgery, the fish were returned to recirculating water heated to 32 °C. Vivo morpholino oligonucleotides (MO) (Gene Tools) containing a 3' fluorescein tag were injected at 1.5 mM into the regenerating tissue on the dorsal part of each zebrafish caudal fin at 48 hpa. The other uninjected half (ventral area) was considered an internal control in order to monitor the normal growth. Immediately after injections, the dorsal half was electropored using a NEPAGENE electroporator, five consecutive 50 msec pulses, at 15 V, with a 30 s pause between pulses, were used. A 3-mm-diameter platinum plate electrode (CUY 650-P3 Tweezers, Protech International) localized the pulses to approximately the dorsal one-half of the caudal fin. The MO sequences are shown in the Reagents and Tools Table.

After the procedure and 48 h post injection (hpi), caudal fins were photographed, capturing fluorescent and brightfield signals simultaneously using a Leica M205 FA fluorescence stereo microscope equipped with a DFC365FX camera (Leica) and LasX software. In order to calculate the percentage area of growth between the injected and non-injected part, the values were inserted in the following formula: (Dorsal 48 hpi - Dorsal 0 hpi)/(Ventral 48 hpi -Ventral 0 hpi)*100, where Dorsal is the regenerative area of the MO-treated tissue and Ventral is the regenerative area of the corresponding uninjected half.

For overexpressing ATRX in the caudal fin regenerating tissue, adult $tert^{-/-}$ zebrafish were anesthetized in 0.05% benzocaine prior to the amputation of the distal portion of their caudal fins, proximal to the first lepidotrichial branching point. Following the surgery, the fish were returned to recirculating water heated to 32 °C. Forty-eight hours later, regenerating tissue was injected with 1 μL of a mix containing 2 μg/μL of pCMV.zfATRX plasmid. Immediately after injections, fish were electropored using a NEPAGENE electroporator. Seven consecutive 50 ms pulses, at 15 V with a 30 s pause between pulses, were used. A 3-mm-diameter platinum plate electrode (CUY 650-P3 Tweezers, Protech International) localized the pulses to the caudal fin.

## Amputation of zebrafish larval caudal fin primordia and chemical exposure by immersion

Forty-eight hours post fertilization (hpf), zebrafish larvae obtained from $tert^{+/-}$ breeding crosses were anesthetized in 0.008% tricaine (Sigma) and placed on an agar plate to amputate the caudal fin primordia with a surgical razor blade just posterior to the end of the notochord. The larvae were individualized in each well of 96-well plates containing ATR Inhibitor IV (CAS 1232410-49-10 Calbiochem) at 10 µM or DMSO in egg water as a negative control. Larval caudal fin was imaged before amputation, after amputation, and 24 hpa to measure and calculate the regeneration area in both conditions.

## Western blot

Forty-eight hours post fertilization zebrafish larvae were treated for 24 h with 10 µM and 50 µM ATR inhibitor IV (Calbiochem) or DMSO as a negative control. The protein extracts were subjected to polyacrylamide gel electrophoresis, wet transferred to a nitrocellulose membrane (Bio-Rad) and analysed by Western blot. The membrane was incubated for 1 h with TTBS (Tween-tris-buffered saline) containing 5% (w/v) skimmed dried milk powder and immunoblotted using phospho-(Ser/Thr) ATM/ATR Substrate Antibody (#2851 Cell Signaling) (dilution 1:1000) for 16 h at 4 °C. The blot was then washed with TTBS and incubated for 1 h at room temperature with the secondary HRP-conjugated antibody (dilution 1:1000) in 5% (w/v) skimmed milk in TTBS. After repeated washes, the signal was detected with the enhanced chemiluminescence reagent and ChemiDoc XRS (Bio-Rad).

# Data availability

No primary datasets have been generated and deposited in this study.

The source data of this paper are collected in the following database record: biostudies:S-SCDT-10_1038-S44319-025-00602-6.

# Peer review information

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

## Acknowledgements

We warmly acknowledge Cynthia Cabello, María C. López-Maya, Inmaculada Fuentes, and Pedro J. Martínez for their excellent technical assistance. This study has been Funded by Fundación Séneca of the Region of Murcia, Spain, FS-10.13039/100007801 (22579/PI/24), Instituto de Salud Carlos III (ISCIII) through the projects PI19/00188 and PI22/00861, and co-funded by the European Union, Fundación Ramón Areces (grant to MLC), University of Murcia (predoctoral contract to EM-B), Universidad Católica de Murcia (predoctoral contract to DH-S), and Consejería de Sanidad de la Región de Murcia (ZEBER contract to JG-C and FA-P). The funders had no role in study design, data collection and analysis, decision to publish, or preparation of the manuscript.

## Author contributions

**Elena Martínez-Balsalobre**: Formal analysis; Validation; Investigation; Visualization; Methodology. **Monique Anchelin**: Formal analysis; Supervision; Validation; Investigation; Visualization; Methodology. **David Hernández-Silva**:

Validation; Investigation; Visualization; Methodology. **Marina C Mione:** Conceptualization; Writing—original draft. **Victoriano Mulero:** Conceptualization; Writing—original draft; Writing—review and editing. **Francisca Alcaraz-Pérez:** Conceptualization; Data curation; Formal analysis; Supervision; Validation; Investigation; Visualization; Methodology; Writing— original draft; Writing—review and editing. **Jesús García-Castillo:** Conceptualization; Data curation; Formal analysis; Supervision; Validation; Investigation; Visualization; Methodology; Writing—original draft; Writing— review and editing. **María L Cayuela:** Conceptualization; Resources; Data curation; Formal analysis; Supervision; Funding acquisition; Validation; Investigation; Visualization; Methodology; Writing—original draft; Project administration; Writing—review and editing.

Source data underlying figure panels in this paper may have individual authorship assigned. Where available, figure panel/source data authorship is listed in the following database record: biostudies:S-SCDT-10_1038-S44319-025-00602-6.

## Disclosure and competing interests statement

The authors declare no competing interests.

# Expanded View Figures

**A**

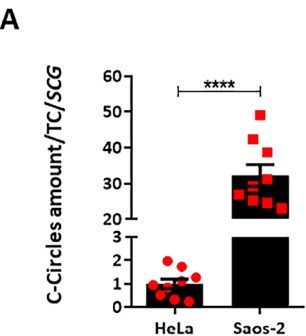

**B**

HeLa 293- C

HeLa 1211-L

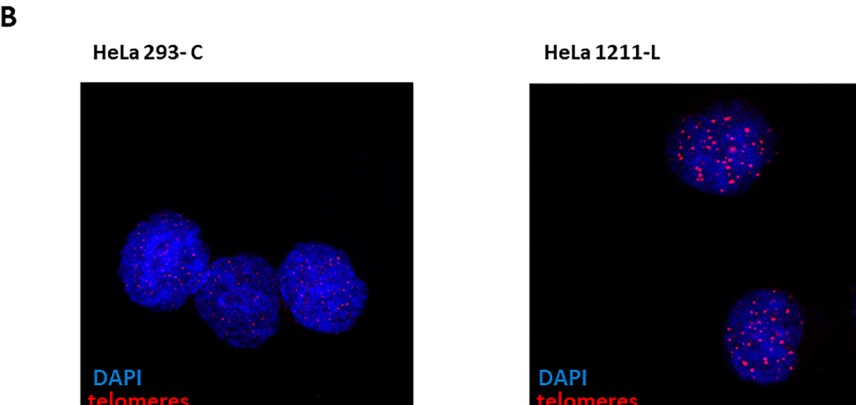

**Figure EV1. C-circles abundance and telomere Q-FISH in control cell lines.**

(**A**) C-circles abundance in ALT-negative human cervical tumor cell line (HeLa) and ALT-positive human osteosarcoma tumor cell line (Saos-2). $n = 3$ experiments with three technical replicates each. (**B**) Q-FISH in HeLa cells. HeLa 293 has a telomere length of 2.7 kb, and HeLa 1211 has a telomere length of 23 kb. Blue=DAPI, red=telomere. Scale bar, 36.8 μm. Data were mean + s.e.m. ****$p < 0,0001$ for the Mann–Whitney test. TC telomere content, SCG single-copy gene. When exact $p$ values are not indicated is because GraphPad Prism software does not show it. Source data are available online for this figure.

**A**

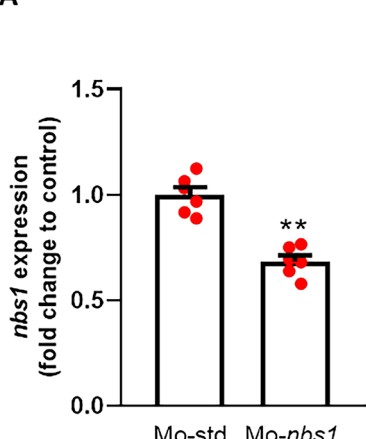

**B**

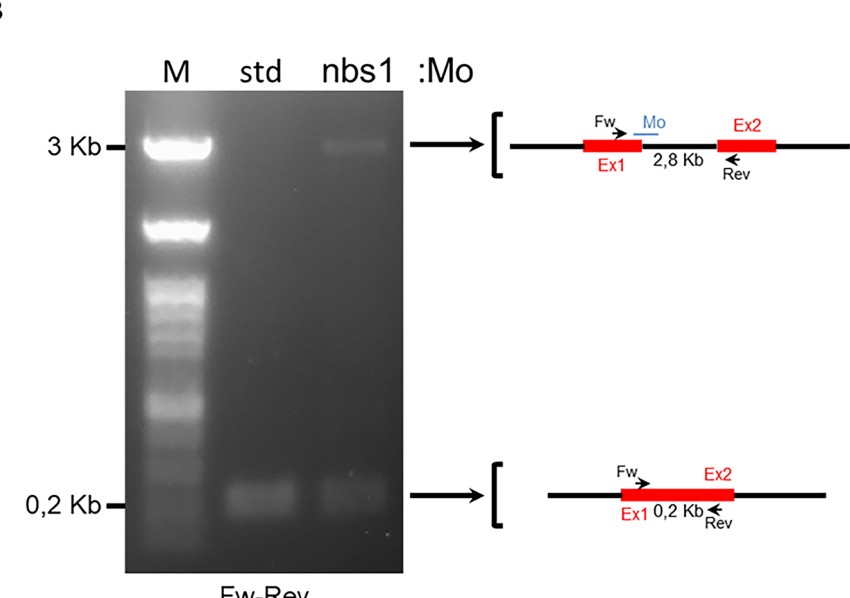

Figure EV2. Efficiency of the nbs1 morpholino.

(A) qPCR showing the expression of nbs1 in 3dpf larvae upon microinjection of standard (Mo-std) or nbs1 (Mo-nbs1) morpholinos. $n = 3$ experiments with two technical replicates each. (B) schematic diagram and PCR showing the retention of the intron 1 of the zebrafish nbs1 gene when the morpholino is used, and the reduction in the amount of the mRNA species splicing the intron. Data were mean ± s.e.m. $p = 0,0022$ for Mann–Whitney test. When exact $p$ values are not indicated is because GraphPad Prism software does not show it. Source data are available online for this figure.

## A

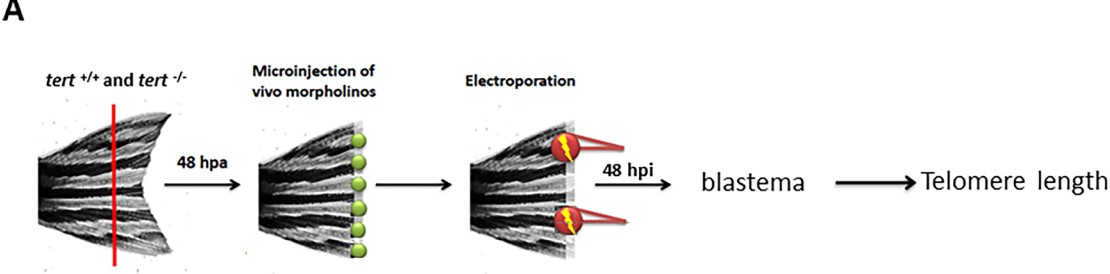

## B

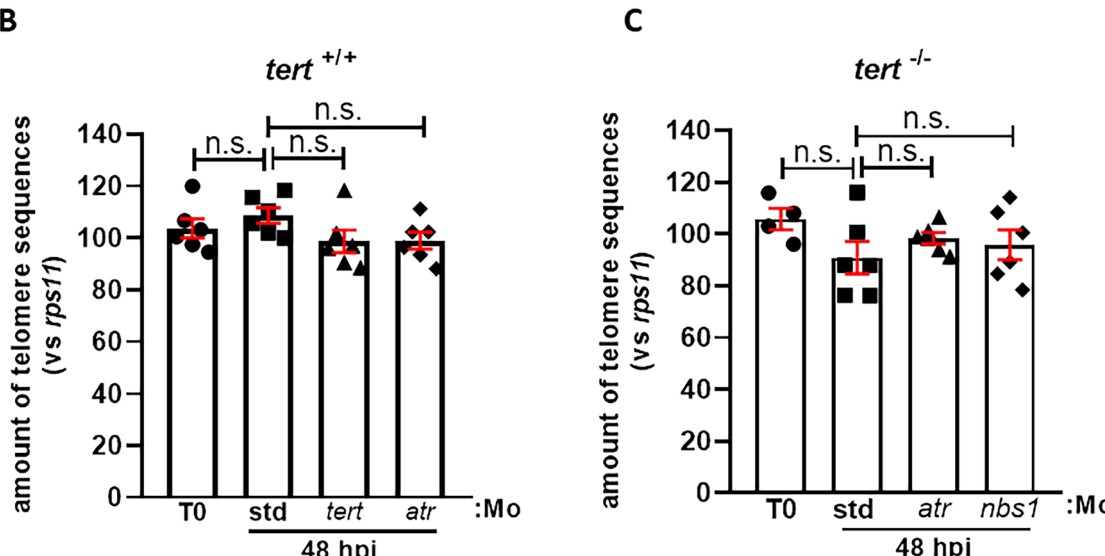

**Figure EV3.  Telomere length is maintained in regenerating tissue of wild-type and tert mutants upon morpholino injection.**

(A) Schematic workflow of the experiment. (B) Quantification by qPCR of telomere length per single-copy gene (rps11) in the regenerated tissue of tert+/+ and (C), tert −/− fish after 48 h post the injection of the indicated morpholinos. T0 is the fin tissue amputated. $n = 3$ experiments. Each dot represents a biological replicate. Data were mean ± s.e.m. n.s. not significant in one-way ANOVA and Bonferroni´s post hoc test. Source data are available online for this figure.

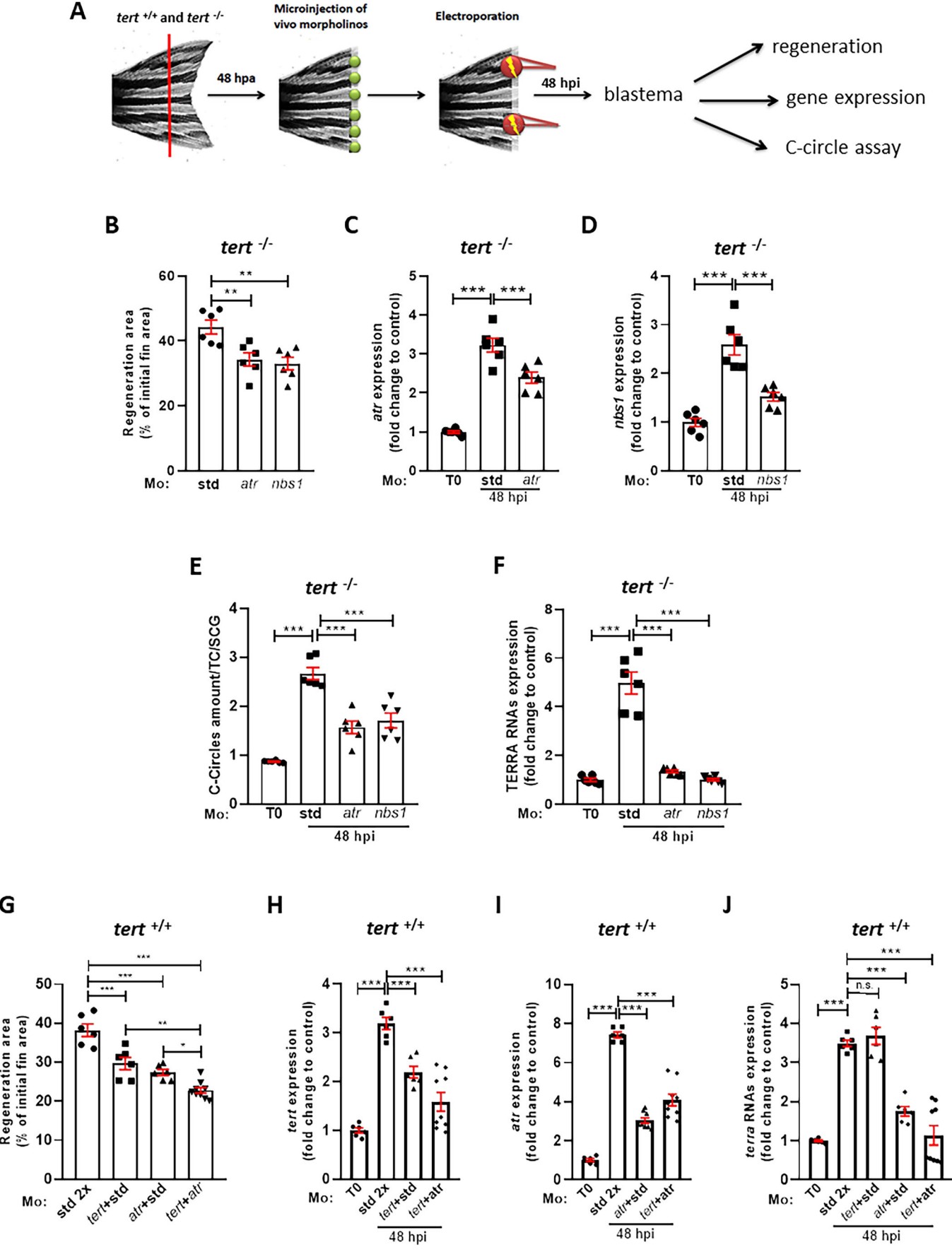

◀ **Figure EV4. Regeneration capacity and ALT mechanism depends on atr and nbs1 in zebrafish caudal fins.**

(A) Schematic diagram of the workflow of the experiment. (B) Regeneration area of caudal fin in tert deficient animals 48 hpa microinjected with the indicated morpholinos. $n = 3$ experiments. Each dot represents a biological replicate. Exact $p$ values are $p = 0.0067$ std vs atr and $p = 0.0027$ std vs nbs. (C, D) qPCRs showing an efficient gene expression downregulation by morpholino injection. $n = 3$ experiments. Each dot represents a biological replicate. Exact $p$ value in (C) is $p = 0.001$ for std vs atr. (E, F) C-circles amount per telomere content (TC) and single-copy gene (SCG) and TERRA RNA levels in regenerated blastema of tert$-/-$ fish microinjected with the indicated morpholinos. $n = 3$ experiments. Each dot represents a biological replicate. (G) Regeneration area of caudal fin in wild-type animals 48 hpa microinjected with double dose of standard (std 2x), tert + std, atr + std, or tert + atr morpholinos. $n = 3$ experiments. Each dot represents a biological replicate. (H–J), qPCRs showing an efficient gene expression downregulation by morpholino injection, and TERRA RNA levels. $n = 3$ experiments. Each dot represents a biological replicate. T0 is the fin tissue amputated. Data were mean ± s.e.m. **$P < 0.01$ and ***$P < 0.001$ for one-way ANOVA plus Bonferroni's post hoc test. n.s. not significant. When exact $p$ values are not indicated is because GraphPad Prism software does not show it. Source data are available online for this figure.

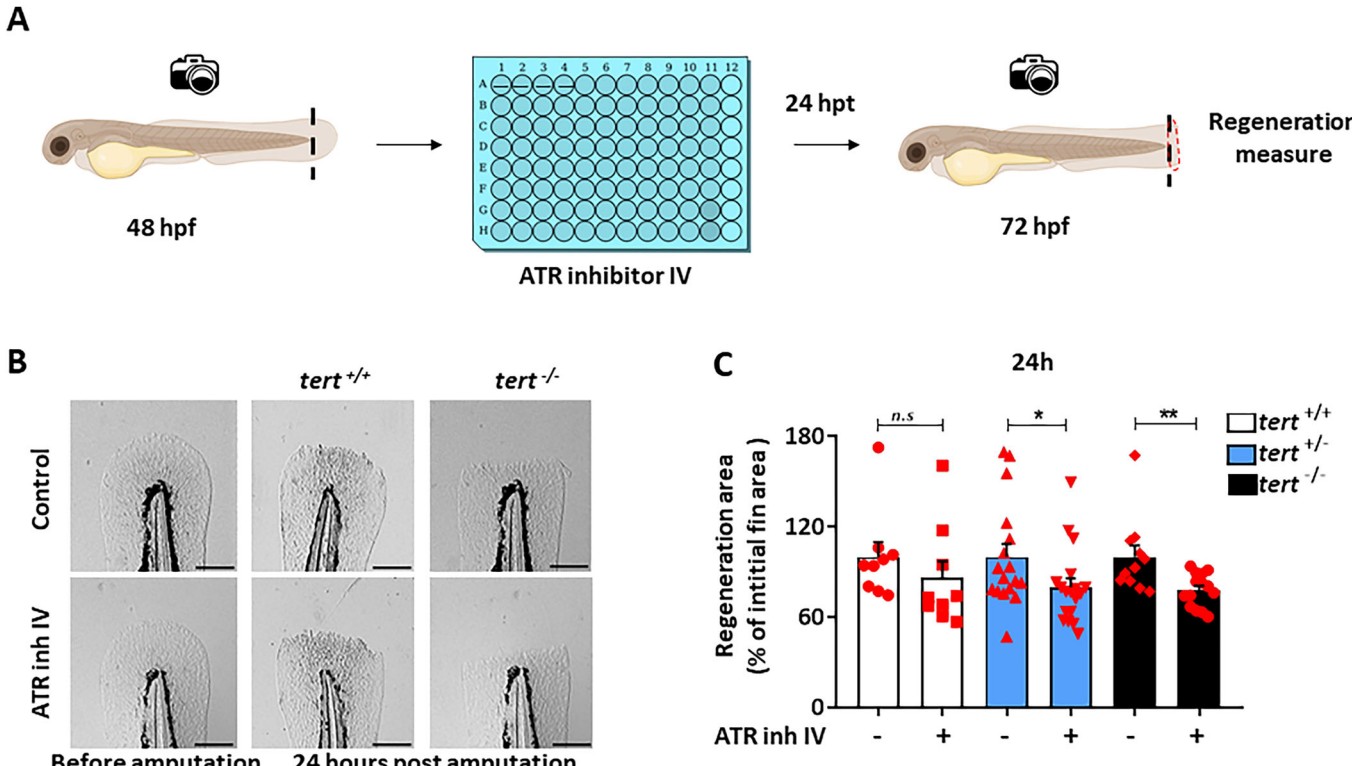

**Figure EV5. Determination of the effect of several concentrations of ATR Inhibitor IV on zebrafish larvae.**

(A) Schematic diagram of the workflow of the experiment. (B) Representative pictures of regenerated caudal fin of the larvae 24 h post amputation treated or not with ATR inhibitor IV. Scale bar, 250 μm. (C) Regeneration of caudal fin of the larvae 24 h post amputation treated or not with ATR inhibitor IV in the three different *tert* genotypes. n = 3 experiments. Each dot represents a biological replicate. Data in (C) is mean + s.e.m. Exact p values are p = 0.042 in *tert*+/− and p = 0.0049 in *tert*−/− for unpaired t-test between treated and untreated. Source data are available online for this figure.

