## [Peer Review File · EMBO Reports]

Telomerase and ALT coexist in the regenerating zebrafish caudal fins

Maria L. Cayuela, Elena Martínez-Balsalobre, Monique Anchin, David Hernandez-Silva, Victoriano Mulero, Marina Mione, Francisca Alcaraz-Pérez, and Jesús García-Castillo

Corresponding author(s): Maria L. Cayuela (marial.cayuela@carm.es), Francisca Alcaraz-Pérez (palcaraz@um.es), Jesús García-Castillo (jesus.garcia@ffis.es)

Review Timeline:

Transfer Date:	12th Jun 25
Editorial Decision:	27th Jun 25
Revision Received:	13th Jul 25
Editorial Decision:	19th Sep 25
Revision Received:	24th Sep 25
Accepted:	30th Sep 25

Editor: Achim Breiling / Esther Schnapp

Transaction Report: This manuscript was transferred to EMBO reports following peer review at Review Commons and another journal.

The logo for Review Commons, featuring the word "Review" in a large, blue, serif font with a diagonal slash through the letter 'v', and the word "COMMONS" in a smaller, blue, sans-serif font below it.

Review #1

1. Evidence, reproducibility and clarity:

Evidence, reproducibility and clarity (Required)

The authors Martínez-Balsalobre and colleagues found that the regenerative capacity of the zebrafish caudal fin is not limited by the lack of telomerase and showed that the length of telomeres does not decrease substantially after repeated amputations in telomerase-deficient zebrafish. These findings prompt the authors to explore an alternative mechanism that would explain the maintenance of telomere length in this regeneration setting. They produced suggestive evidence for the role of the ALT (Alternative Lengthening of Telomeres) mechanism in the maintenance of telomere length in the absence of telomerase in a regeneration setting.

In my view, several points need to be addressed and clarified.

****There are three major points:****

1. When working with tert mutants, the age at which these fish show a telomere phenotype (namely, loss of body mass and reduced fertility) varies. Therefore, it would be important to state if the fish used in this study were already showing these phenotypic characteristics at each time point studied, namely 4, 8 and 11 months of age.

2. The knockdown experiments were performed using morpholinos. To confidently use morpholinos it is fundamental to demonstrate first their knockdown efficiency and their specificity. This is lacking in the manuscript.

3. The involvement of ALT mechanism in the regeneration process in the absence of telomerase is only suggestive, as the authors show an increase of C-circles and heterogenous telomerase length in telomerase-deficient zebrafish but when trying to establish a functional link the authors resort to the knockdown of genes that may be associated with ATL. Looking at the levels of TERRA and the number of C-circles in the knockdown caudal fins would be essential for their claim.

****And several other points:****

4. The regeneration experiments were performed at 32 degrees and this option was never explained nor discussed.

5. When referring to the ALT mechanism, the authors state that "... in about 10% of tumors cells, telomere length is maintained by the Alternative Lengthening of Telomeres (ALT) mechanism ..." and I think it would be more accurate to talk about cancer cells instead of tumor cells.

6. The sentence about C-circles is incorrect. C-circles are mostly single-stranded and not double-stranded as stated.

7. After Figure 2, the authors never mention the age of the fish used.

8. In Figure 1A

The site of amputation does not fit the one described in Mat & Met that states 2 cm from the base of the caudal peduncle. The same stands for Figure 2A.

The experimental procedure refers 1 dpa but this time point is not plotted in the graph in Figure 1B.

9. In Figure 1B

The Y axis should be named regeneration area instead of rate as the values are a percentage of the area reached after a certain time point after amputation. The same stands for Figure 2B, C.

It would be nice to see the real caudal fin images for the relevant time points: before amputation, 0 dpa, when the fins reach 50% of regeneration area and then the last time point.

The authors should discuss why are the caudal fins reaching more than 100% of regeneration area.

10. In Figure 2B

The meaning of "... ." on the right side of the graph is not clear. The same stands for Figure 2D.

11. In Figure 2C

Why is the clip 10, 11 and 12 missing from the tert+/- and tert-/- ?

12. In Figure 2E

The proximity of all points at the 12 Clip is indicative of lack of statistical significance, therefore the **** related to which comparisons?

13. In Figure 2D, E

For the measurement of telomere length, the authors state that "Data are average of at least 2 independent experiments." What does this mean exactly? How many animals were used in each experiment?

14. In Figure 3

The authors state that "Data are average of at least 2 independent experiments." What does this mean exactly? How many animals were used in each experiment?

Why were the c-circles evaluated at hpa while the telomere length evaluated at dpa? This should be discussed.

15. In Figure 3A

The meaning of ". . ." on the top side of the graph is not clear.

t0 should be removed and replaced by 0 hpa and 24hpa and 48hpa for coherence.

16. In Figure 3B,C

0 hpa replace by 0 dpa

17. In Figure 3B

The blue and red stainings in the panels are labelling exactly what? This should be stated in the image and in the legend.

18. In Figure 3D

There is a mistake in the legend the should be corrected as follows "Very long telomeres have a higher fluorescence of 200,000 AUF and very short telomeres have a lower fluorescence of 30,000 AUF."

19. In Figure 4

t0 should be removed and replaced by 0 hpa.

The meaning of ". . ." on the top side of the graph is not clear.

The title is an overstatement, as the genes studied are DNA damage genes that may associate with ALT.

20. In Figure 4A, B

The expression of nbs1 and atr in tert^{-/-} increases at 48hpa but the same seems to be true for the tert^{+/+} and this is never discussed by the authors.

21. In Figure 4C, D

The differences in the expression of atrx and daxx decreases over time in a in tert^{-/-} and this

is never discussed by the authors.

22. In Figure 5

An ideal control would be the direct comparison between microinjected+electroporated mo-std in the ventral part of the fin while the dorsal part would be microinjected+electroporated with the mo-gene of interest. This would discard any effect of microinjection+electroporation in the regeneration efficiency.

These experiments are not convincing to show that there is an ALT mechanism is operating here. What this experiment shows is the relevance of these genes for the regenerative capacity of the caudal fin. To show that this is related to the ALT mechanism the authors should investigate the C-circles in these regenerating fins.

The amputation red lines are not placed in the exact amputation position in some of the panels.

Regeneration rate should be regeneration area.

23. In Figure 5C, E

Why is the mo-tert more inhibitory of regeneration (Figure 5E - around 30%) than the tert-/- mutant (Figure 5C - around 60%)? This should be discussed.

24. In Figure 6A

The 2 adult zebrafish shown in the tank with the ATR inhibitor IV should have an amputated caudal fin.

Control is exactly what? Untreated? Treated with vehicle?

Why was the ATR inhibitor IY added immediately after fin amputation while the mo-atr was injected at 48 hpa?

25. Figure 6D, E, F

These panels are a bit out of the focus of this paper. If presented should go to a supplementary figure.

26. In Figure S2

The relevant bands should be identified.

The gel identifies DMSO, 10uM and 50 uM but the quantification graph identifies Control, 50uM and 100uM.

There are no error bars.

The authors say that the quantification of various western blot bands was done but how many exactly?

27. In Figure S3

The primers for rps11 are repeated twice.

Were these primers designed de novo by the authors or did they use previously reported primers, in this case the references should be given.

Tert F2 and R1 should be replaced for F and R for consistency.

28. In Figure S4

The sequence of tert mo is missing.

29. In the methods the genotyping protocol of tert mutants is not described.

30. The method to calculate the area of the fin pre- and post-amputation is not described.

2. Significance:

Significance (Required)

The manuscript by Martínez-Balsalobre and colleagues deals with a very interesting question on the importance of telomere lengthening during regenerative processes and its relation to ageing. To this end the authors made use of the tert mutant, a telomerase-deficient zebrafish. The authors show a surprising phenotype that telomerase-deficient zebrafish can still regenerate their caudal fins and are able to maintain telomere length during consecutive amputations and I say surprising because it has been shown that telomerase-deficient zebrafish are unable to regenerate their hearts efficiently.

Taking these novel findings, the authors propose that in the zebrafish caudal fin and in the absence of telomerase, telomere length is maintained through the activation of an alternative mechanism called ALT. To my knowledge, the role of ALT as a mechanism of telomere lengthening has never been described in the context of regenerating organs in zebrafish.

****Referees cross-commenting****

I agree with the comments made by the other reviewers. I would stress the need to tone down the role of ALT during fin regeneration in zebrafish as all the experiments are only indicative of the possibility of the involvement of ALT.

3. How much time do you estimate the authors will need to complete the suggested revisions:

Estimated time to Complete Revisions (Required)

(Decision Recommendation)

More than 6 months

Review #2

1. Evidence, reproducibility and clarity:

Evidence, reproducibility and clarity (Required)

Using zebrafish as a model for regeneration, the authors find that telomere maintenance by recombination can occur in the absence of telomerase.

Title to Figure 4 perhaps may be too strong, 'ALT mechanism is activated', since only a few features of ALT are assessed. Perhaps, 'ALT features are activated'?

mRNA levels of NBS, ATR are also increased in WT animals (Figure 4A and 4B), but ATRX and DAXX mRNA levels are not decreased in WT animals. Is the increase why the authors in part suggest that ALT is being used in WT animals. If so, what would be the trigger for the use of ALT, as opposed to the trigger to use ALT in *tert*^{-/-} animals?

In Figure 5C, if *tert*^{-/-} animals are downregulated for *nbs1* and *atr*, would it be expected that the effect on regeneration be more pronounced compared to *tert*^{+/+} downregulated for *nbs1* and *atr* than what is observed?

What are the telomere lengths in *tert*^{-/-} animals treated with *mo-atr* or *mo-nbs1* or in *tert*^{+/+} animals treated with *mo-tert* and *mo-atr* compared to singly treated?

2. Significance:

Significance (Required)

Reported findings are novel, timely and model of possible therapeutic value for screening compounds for ALT and/or telomerase inhibitors. Mechanisms of co-existence of ALT and telomerase can also be explored using this model.

3. How much time do you estimate the authors will need to complete the suggested revisions:

Estimated time to Complete Revisions (Required)

(Decision Recommendation)

Between 1 and 3 months

Review #3

1. Evidence, reproducibility and clarity:

Evidence, reproducibility and clarity (Required)

****Summary:****

Martinez-Balsalobre have examined caudal fin regeneration following surgical transection in WT and telomerase-deficient (*tert*^{+/-} and *tert*^{-/-}) zebrafish adults of several ages, and in one experiment, in embryos. They conclude: (1) regeneration efficiency decrease with aging in all genotypes (2) telomere length is maintained, even in a *tert*-mutant background (3) ALT (alternative lengthening of telomeres) is involved in supporting cell proliferation in tailfin regeneration. The experimental system employs a quantitative area-based measurement as a measure of the degree of regeneration. Functional studies used antisense morpholino gene knockdown and chemical inhibition to implicate ALT involvement.

****Major comments:****

The experimental logic is appropriate, and in general, the data support the conclusions. Strengths of the work include: (1) The quantitative measure of % regeneration appears to be quite objective; (2) the internally controlled experimental design of the morpholino knockdown experiments of Fig 5.

The Western blot in Fig S2 has some issues. The image is a montage. The experiment appears to have been done only once. The band's identifications by kDa are imprecise (where is the 82 kDa band on the gel? - there are bands smaller and larger than 82 kDa, but none of 82kDa; the 50 kDa band is close to background; the DMSO lane is underloaded relative to the two test lanes (but as the observation is a reduction in the test samples, this does not result in a misinterpretation). What concentration of ATRinhIV were used? The blot has 10 and 50 microM, Fig S1B has 50 and 100 microM, and the text says 1-50 microM).

The MO-knockdown studies are interpreted as showing synergy of atr and tert knockdown. There are two problems with their interpretation of synergy: (1) the single result of a greater effect with both MOs does not distinguish between an additive or synergistic effect (and synergistic action is by definition a greater than additive action); (2) MO dose is not controlled by a group with an equal total MO doses (mo-std+mo-atr and mo-std+mo-tert). While acknowledging that the issues of using local MO delivery in an adult model are very different from global delivery in an embryonic model, the "synergy" interpretation still requires these experiments/controls be done. These experiments were not accompanied by any molecular evidence that either of the morpholinos targeted expression of the intended gene (which would likely have to be derived from their assessment in another system) - a control that can be challenging, but one that is regarded as essential in the field (<https://doi.org/10.1242/dev.001115>). While this will be difficult to do in the adult setting, it is still appropriate to validate the activity/molecular efficacy of the MO sequence in an experimentally tractable scenario. The specificity of this experiment and interpretation would also be enhanced and corroborated independently by undertaking the atr knockdown in the tert -/- mutant background. Overall, these experiments were preliminary and require further work that could be done within 3 months.

Note - the tert MO sequence is missing from the table in Fig S4.

The adult experiments have used n=6-10 animals/group. There is no consideration of statistical power (is the analysis of Fig 1C adequately powered?).

The degree and nature of replication is not clear in all cases. For example, in Fig 1, were the 6 fish run as one cohort of 6 animals in parallel (which would be just one experiment with 6 animals, each animal being a biological replicate), or were there 6 animals injured at different times (representing multiple independent experiments and represented a greater degree of reproducibility), or something in between. A similar question applies to the other figures.

For the experiment of Fig 6F, although there are ≥ 100 larvae per group, it is not clear that this experiment has been done more than once.

A few comments about data presentation. "Regeneration rate" and its derivatives are presented as mean \pm SEM. The parameter measured is correctly defined in methods as "Percent fin regeneration", however the graphs where it is plotted have the y-axis labelled as "regeneration rate (%)" (for example, Fig 1B), which is incorrect. The plotted parameter is not a rate - although there is a time dimension (x-axis), what is plotted at each time point is "% regeneration". Also, in most figures, such as Fig 1B and 1C, mean \pm SD would be more

appropriate, as here each of the n=6 data points represents a single observation from one individual in the population, not the mean of 6 small samples of groups of individuals from the population. Furthermore, at these small n-values (6-10 through the report), scatter plots are considered a more appropriate way of displaying the data (some succinct references: DOI: 10.4103/2229-3485.100662 ; from a Nature group journal DOI: 10.1038/s41551-017-0079 ; from a PLOS journal <https://doi.org/10.1371/journal.pbio.1002128>). The use of mean +/- SEM in Fig 4 could be appropriate, but as n is "at least two independent experiments" scatter plots would again be appropriate. Readers would then know which data sets had only two values. In two instances, the same data are presented in two different ways (Fig 1B, 1C; the column graphs and arrows of Fig 3D).

How does "data are average of at least 2 independent experiments" apply to Fig 3C?

****Minor comments:****

The paper is written clearly overall. There are multiple minor grammatical/typographical errors, but these did not detract from understanding the manuscript. These were most abundant in the discussion.

A few points:

Discussion p1 - what is meant by "prematurely aged 11-month fish"

Discussion p2 - you mean "doubled" rather than duplicated?

tert +/+, tert +/- and tert -/- genotypes for experiments - how were these obtained and genotypically verified? (heterozygous incrosses? WT x homozygous mutant outcrosses?)

The last paragraph of the discussion makes some valid points, but it seemed out of place and I wondered if it was misplaced.

The rps11 primers appear in the Table of Fig S3 twice.

2. Significance:

Significance (Required)

The authors claim that this is the first in vivo model examining ALT in regeneration.

The paper contributes to the relatively small body of literature using adult zebrafish models (rather than embryonic larval models) in biomedical research. I cannot comment on the telomere/telomerase literature.

This report will be of interest to those working in regenerative medicine, telomere biology, cancer research, and those interested in zebrafish models of disease and physiological

processes.

My expertise encompasses zebrafish disease models and functional studies; I do not have special expertise in telomerase or ALT pathways.

3. How much time do you estimate the authors will need to complete the suggested revisions:

Estimated time to Complete Revisions (Required)

(Decision Recommendation)

Between 1 and 3 months

Full Revision

Manuscript number: RC-2021- 01152R

Corresponding author(s): MARIA L. CAYUELA

1. General Statements [optional]

This section is optional. Insert here any general statements you wish to make about the goal of the study or about the reviews.

This section is mandatory. Please insert a point-by-point reply describing the revisions that were already carried out and included in the transferred manuscript.

Reviewer #1 (Evidence, reproducibility and clarity (Required)):

The authors Martiñez-Balsalobre and colleagues found that the regenerative capacity of the zebrafish caudal fin is not limited by the lack of telomerase and showed that the length of telomeres does not decrease substantially after repeated amputations in telomerase-deficient zebrafish. These findings prompt the authors to explore an alternative mechanism that would explain the maintenance of telomere length in this regeneration setting. They produced suggestive evidence for the role of the ALT (Alternative Lengthening of Telomeres) mechanism in the maintenance of telomere length in the absence of telomerase in a regeneration setting. In my view, several points need to be addressed and clarified.

****There are three major points:****

1. When working with *tert* mutants, the age at which these fish show a telomere phenotype (namely, loss of body mass and reduced fertility) varies. Therefore, it would be important to state if the fish used in this study were already showing these phenotypic characteristics at each time point studied, namely 4, 8 and 11 months of age

The premature aging phenotype of *tert* mutant fish has been previously characterized in the paper by Anchin et al 2013 referenced in the manuscript. We used young fish with no phenotype (4 months old), and aged fish (8 and 11 months old) presenting the already described premature aging phenotypes, such as spinal curvature, loss of fertility, loss of body mass and loss of pigmentation.

The following sentence regarding this has been included in the revised version of the manuscript. "The fish used showed non-detectable aging phenotype at 4 months old, whereas at 8- and 11-months fish presented the typical *tert* mutant premature aging phenotypes, i.e. backbone curvature, loss of body mass and hypopigmentation"

2.The knockdown experiments were performed using morpholinos. To confidently use morpholinos it is fundamental to demonstrate first their knockdown efficiency and their specificity. This is lacking in the manuscript.

In this work we have used 3 different morpholinos; *tert* morpholino has been already used and characterized in the work by Imamura and collaborators in 2008. *atr* morpholino has been already used and characterized in the paper by Stern et al., 2005.

However, *nbs1* morpholino has been designed for this work. A Supplemental Figure (Figure S2) and the following paragraph have been added in the revised version of the manuscript to show the knock-down efficiency of the *nbs1* morpholino:

“The knock-down efficiency of the *atr* morpholino was characterized by Stern and colleagues (Stern et al, 2005). The injection of the *nbs1* morpholino in zebrafish eggs resulted in the reduction of the expression of *nbs1* mRNA at 3dpf (Fig. S2A). Furthermore, PCR using cDNA as a template detected *nbs1* mRNA species that retained the intron one of the gene as a result of the morpholino effect in blocking the splicing (Fig. S2B).

“The *tert* morpholino knock-down efficiency has been already showed (Imamura et al, 2008)”

3.The involvement of ALT mechanism in the regeneration process in the absence of telomerase is only suggestive, as the authors show an increase of C-circles and heterogenous telomerase length in telomerase-deficient zebrafish but when trying to establish a functional link the authors resort to the knowndown of genes that may be associated with ATL. Looking at the levels of TERRA and the number of C-circles in the knowndown caudal fins would be essential for their claim.

We have now performed caudal fin regeneration experiments in *tert* mutant fish microinjected with *mo-atr* and *mo-nbs1* and analyzed the levels of TERRA RNAs and C-circles amount. The results are shown in Supplemental Figure S4. As expected, regeneration capacity decreased in fish microionjected with both morpholinos compared to control fish (FigS4 F). Consistently, TERRA RNAs levels, as well as C-circles amount, increased in the regenerating tissue and this induction was lower when *atr* and *nbs1* gene expression was downregulated by *mo*-injection (Fig S4 G-J). Taking altogether, these results indicate that ALT mechanism is induced upon amputation and operates in the regenerating tissue of *tert* deficient fish.

And several other points:

4.The regeneration experiments were performed at 32 degrees and this option was never explained nor discussed.

The regeneration experiments in zebrafish typically are performed at 32 °C to accelerate regeneration process. Otherwise, the amount of regenerated blastema at 48 hpa or 72hpa would not be enough to perform any kind of analysis. Furthermore, it could happen that some experimental modifications, for instance the effects of the morpholino injection, do not last if the regeneration process is kept more than 84-96hpa at 28 °C.

Full Revision

This procedure have been used previously by other laboratories (PMID: 8601496, Johnson and Weston, 1995; PMID: 12015289 Nechiporuk et al., 2003 and PMID: 16273523 Thumnel et al 2006) to increase the rate of regeneration approximately two fold, a temperature of 33°C was used for the regeneration experiments. In addition, It has been demonstrated normal regeneration at 33°C in wild-type fish

5. When referring to the ALT mechanism, the authors state that "... in about 10% of tumors cells, telomere length is maintained by the Alternative Lengthening of Telomeres (ALT) mechanism ..." and I think it would be more accurate to talk about cancer cells instead of tumor cells.

This has been corrected in the revised version

6. The sentence about C-circles is incorrect. C-circles are mostly single-stranded and not double-stranded as stated.

This has been corrected in the revised version

7. After Figure 2, the authors never mention the age of the fish used.

All the fish used in the amputation experiments after Fig2 are 4 -6 months of age

8. In Figure 1A. The site of amputation does not fit the one described in Mat & Met that states 2 cm from the base of the caudal peduncle. The same stands for Figure 2A.

This is corrected in the new version with a new Figure 1A and 2A

9. In Figure 1B

The Y axis should be named regeneration area instead of rate as the values are a percentage of the area reached after a certain time point after amputation. The same stands for Figure 2B, C. It would be nice to see the real caudal fin images for the relevant time points: before amputation, 0 dpa, when the fins reach 50% of regeneration area and then the last time point.

This has been changed in the new version

The authors should discuss why are the caudal fins reaching more than 100% of regeneration are

This is an intriguing question for which we currently lack an answer. Nonetheless, it does not impact the focus of our ongoing study

10. In Figure 2B. The meaning of "." on the right side of the graph is not clear. The same stands for Figure 2D.

This has been a mistake when handling the figure folder and has been corrected in the revised version

11. In Figure 2C .Why is the clip 10, 11 and 12 missing from the tert+/- and tert-/- ?

This has been changed in the new version and recalculated the statistical significance. We appreciate the feedback

12. In Figure 2E The proximity of all points at the 12 Clip is indicative of lack of statistical significance, therefore the **** related to which comparisons?

We have modified the data of fig 2E and recalculated the statistical significance

13. In Figure 2D, E

For the measurement of telomere length, the authors state that "Data are average of at least 2 independent experiments." What does this mean exactly? How many animals were used in each experiment?

In the experiments in Fig2, 6 fish total were used per group sampled in at least 2 independent experiments. This has been included in the figure legend and in the Mat&Met section

14. In Figure 3

The authors state that "Data are average of at least 2 independent experiments." What does this mean exactly? How many animals were used in each experiment?

The experiment in Fig 3A was done 3 times with 2 fish per group pooled in each experiment. The telomere length experiment has been done 2 times. This has been added to the figure legend and to the Mat&Met section.

Why were the c-circles evaluated at hpa while the telomere length evaluated at dpa? This should be discussed.

We expect to observe an effect on telomere length after several days of continuous cell proliferation in order to completely regenerate the caudal fin. However, the presence of C-circles in the regenerating tissue is expected to be found as early as 24hpa as a consequence of the action of the ALT mechanism of telomere maintenance, which has to be active from the very beginning. The following sentence has been included in the Discussion section: "ALT activation is expected to happen, and in fact detected, very early in the regeneration process, and eventually results in telomere length heterogeneity several days after amputation, when a lot of cell divisions and telomere recombination have occurred".

15. In Figure 3A

The meaning of ". . ." on the top side of the graph is not clear.

t0 should be removed and replaced by 0 hpa and 24hpa and 48hpa for coherence.

This has been a mistake when handling the figure folder and has been corrected in the revised version.

16. In Figure 3B,C 0 hpa replace by 0 dpa

This has been replaced in the new version

17. In Figure 3B

The blue and red stainings in the panels are labelling exactly what? This should be stated in the image and in the legend.

Red staining represents the telomeres and the blue staining are the nuclei. It is shown in the Figure and stated in the figure legend.

18. In Figure 3D

Full Revision

There is a mistake in the legend the should be corrected as follows "Very long telomeres have a higher fluorescence of 200,000 AUF and very short telomeres have a lower fluorescence of 30,000 AUF."

This has been corrected

19. In Figure 4

t0 should be removed and replaced by 0 hpa.

This has been corrected

The meaning of ". . ." on the top side of the graph is not clear.

This has been a mistake when handling the figure folder and has been corrected in the revised version

The title is an overstatement, as the genes studied are DNA damage genes that may associate with ALT.

The title has been corrected to "The expression of ALT-associated genes is modulated in regenerative tissue of"

20. In Figure 4A, B

The expression of *nbs1* and *atr* in *tert*^{-/-} increases at 48hpa but the same seems to be true for the *tert*^{+/+} and this is never discussed by the authors.

This result would support the idea that both telomerase-dependent and ALT mechanisms operate in the regeneration process in a wild type animal. A sentence in the results and discussion sections has been added to mention and discuss this point:

"These genes were quantified in the regenerated tissue at 24 and 48 hpa. *nbs1* and *atr* mRNA levels increased in telomerase deficient fish at 48 hpa compared to time 0 (0hpa) (Fig. 4A, 4B).

The same effect in the expression of these genes was found in wild type fish regenerating fins.

Interestingly, *atrx* and *daxx* expression decreased (Fig. 4C, 4D) at 24 and 48 hpa, in agreement with published data on ALT in cells (Amorim et al., 2016; Ren et al., 2018; Yost et al., 2019)."

"Curiously we observed an increased expression of ALT activator proteins in both wild type and telomerase deficient zebrafish, and a decrease in ALT inhibitor proteins suggesting that the main players of ALT and their mechanisms are conserved during evolution, and that both mechanisms of telomere maintenance could co-exist in the regeneration process in wild type fish"..

21. In Figure 4C, D

The differences in the expression of *atrx* and *daxx* decreases over time in a *tert*^{-/-} and this is never discussed by the authors.

As mentioned, and referenced in the manuscript, the proteins are ALT inhibitors, and mutations in these proteins are described to be promoting the activation of ALT mechanisms. Thus, it is expected that in the regenerating fins where ALT is activated, their expression decreases.

22. In Figure 5

An ideal control would be the direct comparison between microinjected+electroporated *mo*-std in the ventral part of the fin while the dorsal part would be microinjected+electroporated with the *mo*-gene of interest. This would discard any effect of microinjection+electroporation in the regeneration efficiency.

These experiments are not convincing to show that there is an ALT mechanism is operating here. What this experiment shows if the relevance of these genes for the regenerative capacity of the

caudal fin. To show that this is related to the ALT mechanism the authors should investigate the C-circles in these regenerating fins.

We have performed regeneration experiments using WT fish to address this issue. We analyzed the regenerated area of control and morpholino injected fish and then obtained regenerating blastema and analyzed the expression of *tert* and *atr*. The results are shown in Supplemental Figure S4 (A-E). The regeneration capacity is inhibited in tissues injected with a mix of mo-std+mo-ter, a mix of mo-std+mo-atr, or a mix of mo-tert+mo-atr compared with a control injected with a double dosis of mo-std (std 2x, Fig S4B). In addition, the expression of *tert* and *atr* is decreased in the regenerated blastema upon morpholino injection (Fig S4 C and D) indicating that the genetic inhibition of the expression of these genes was efficient. Finally, the levels of TERRA RNAs are increased upon amputation and this induction is reduced when we mo-atr or a combination of mo-atr+tert were microinjected (Fig S4E).

We have also performed caudal fin regeneration experiments in *tert* mutant fish microinjected with mo-atr and mo-nbs1 and analyzed the levels of TERRA RNAs and C-circles amount. The results are shown in Supplemental Figure S4. As expected, regeneration capacity of the caudal fins of fish microinjected with both morpholinos decreased compared to control fish (Fig S4 F-H). Consistently, TERRA RNAs levels, as well as C-circles amount, increased in the regenerating tissue and this induction was lower when *atr* and *nbs1* gene expression was decreased by mo-injection (Fig S4 I and J). Taking altogether, these results indicate that ALT mechanism is induced upon an injury and operating in the regenerating tissue of both wild type and *tert* deficient fish.

The amputation red lines are not placed in the exact amputation position in some of the panels. Regeneration rate should be regeneration area.

This has been corrected

23. In Figure 5C, E

Why is the mo-tert more inhibitory of regeneration (Figure 5E - around 30%) than the *tert*^{-/-} mutant (Figure 5C - around 60%)? This should be discussed.

This point is now discussed: “

24. In Figure 6A

The 2 adult zebrafish shown in the tank with the ATR inhibitor IV should have an amputated caudal fin.

This has been modified

Control is exactly what? Untreated? Treated with vehicle?

The control is fish treated with the same amount of DMSO (vehicle). This is now shown in the panel

Why was the ATR inhibitor IV added immediately after fin amputation while the mo-atr was injected at 48 hpa?

The ATR inhibitor was added immediately after amputation because ALT is then inhibited from the starting of the regeneration process. However, in the case of the *atr* morpholino we need some regenerated tissue to perform the microinjection within and inhibit *atr* expression specifically in this tissue.

25. Figure 6D, E, F

These panels are a bit out of the focus of this paper. If presented should go to a supplementary figure.

These panels are now moved to the Supplemental Figure S5

26. In Figure S2

The relevant bands should be identified.

We have performed new regeneration experiments in wild type adult fish using ATR inhibitor. The results show that treating fish with ATR inhibitor provokes a clear decrease in the overall phosphorylation status of ATR/ATR substrates within the regenerated tissue (Figure 5B and C). In this case, the intensity of the whole lane was used for quantification.

The gel identifies DMSO, 10uM and 50 uM but the quantification graph identifies Control, 50uM and 100uM.

This has been corrected in the new version

There are no error bars

In the new experiments are now shown.

The authors say that the quantification of various western blot bands was done but how many exactly?

In the new experiments, 3 western blots are quantified

27. In Figure S3

The primers for rps11 are repeated twice. Were these primers design de novo by the authors or did they used previous reported primers, in this case the references should be given.

Tert F2 and R1 should be replaced for F and R for consistency.

This has been corrected and references for the primers used are added in the new Supplemental Figure S6

28. In Figure S4

The sequence of tert mo is missing.

This has been corrected

29. In the methods the genotyping protocol of tert mutants is not described.

A protocol for genotyping the tert deficient zebrafish has been added in the Mat&Met section.

30. The method to calculate the area of the fin pre- and post-amputation is not described.

The method is already described in the Mat&Met section: "In order to calculate the percentage area of growth between the injected and non-injected part, the values were inserted in the following formula: $(\text{Dorsal } 48 \text{ hpi} - \text{Dorsal } 0 \text{ hpi}) / (\text{Ventral } 48 \text{ hpi} - \text{Ventral } 0 \text{ hpi}) * 100$, where Dorsal is the regenerative area of the MO-treated tissue and Ventral is the regenerative area of the corresponding uninjected half"

Reviewer #1 (Significance (Required)):

The manuscript by Martiñez-Balsalobre and colleagues deals with a very interesting question on the importance of telomere lengthening during regenerative processes and its relation to ageing. To this end the authors made use of the tert mutant, a telomerase-deficient zebrafish. The authors show a surprising phenotype that telomerase-deficient zebrafish can still regenerate their caudal fins and are able to maintain telomere length during consecutive amputations and I say surprising because it has been shown that telomerase-deficient zebrafish are unable to regenerate their hearts efficiently.

Taking these novel findings, the authors propose that in the zebrafish caudal fin and in the absence of telomerase, telomere length is maintained through the activation of an alternative mechanism called ALT. To my knowledge, the role of ALT as a mechanism of telomere lengthening has never been described in the context of regenerating organs in zebrafish.

We fully appreciate the reviewer's comments on the significance of the manuscript!

****Referees cross-commenting****

I agree with the comments made by the other reviewers. I would stress the need to tone down the role of ALT during fin regeneration in zebrafish as all the experiments are only indicative of the possible of the involvement of ALT.

We have conducted additional experiments that further support the involvement of ALT. Please read the responses to the other reviewers for more details.

Reviewer #2 (Evidence, reproducibility and clarity (Required)):

Using zebrafish as a model for regeneration, the authors find that telomere maintenance by recombination can occur in the absence of telomerase.

Title to Figure 4 perhaps may be too strong, 'ALT mechanism is activated', since only a few features of ALT are assessed. Perhaps, 'ALT features are activated'?

The title to Figure 4 has been changed to "The expression of ALT-related genes is modulated..." mRNA levels of NBS, ATR are also increased in WT animals (Figure 4A and 4B), but ATRX and DAXX mRNA levels are not decreased in WT animals. Is the increase why the authors in part suggest that ALT is being used in WT animals. If so, what would be the trigger for the use of ALT, as opposed to the trigger to use ALT in *tert*^{-/-} animals?

Our results indicate the utilization of both telomere maintenance mechanisms to support cell division in regenerative fins among wild-type animals. Consequently, we propose that the signals instigating regeneration are shared between both mechanisms and are present in both wild-type and *tert*-deficient animals, albeit with varying degrees of contribution.

In Figure 5C, if *tert*^{-/-} animals are downregulated for *nbs1* and *atr*, would it be expected that the effect on regeneration be more pronounced compared to *tert*^{+/+} downregulated for *nbs1* and *atr* than what is observed?

We agree with the reviewer comment, and that is what actually happens. The inhibition of the regeneration in wild type fish is about 40% in *mo-nbs1* injection and around 70% in *mo-atr* injected animals. However, in *tert* mutants, the decrease in regeneration observed in *mo-nbs1* injection is about 56%, whereas is 82% in *mo-atr* injection.

What are the telomere lengths in *tert*^{-/-} animals treated with *mo-atr* or *mo-nbs1* or in *tert*^{+/+} animals treated with *mo-tert* and *mo-atr* compared to singly treated?

The telomere length does not change in *mo-atr* or *mo-nbs1* injected *tert* mutants compared to *mo-std* animals.

The telomere length in mo-tert and mo-atr injected wild type animals does not change compared to mo-std injected animals.

This results are now shown in Supplemental Figure S3

Reviewer #2 (Significance (Required)):

Reported findings are novel, timely and model of possible therapeutic value for screening compounds for ALT and/or telomerase inhibitors. Mechanisms of co-existence of ALT and telomerase can also be explored using this model.

We fully appreciate the reviewer's comments on the significance of the manuscript!

Reviewer #3 (Evidence, reproducibility and clarity (Required)):

****Summary:****

Martinez-Balsalobre have examined caudal fin regeneration following surgical transection in WT and telomerase-deficient (tert+/- and tert-/-) zebrafish adults of several ages, and in one experiment, in embryos. They conclude: (1) regeneration efficiency decrease with aging in all genotypes (2) telomere length is maintained, even in a tert-mutant background (3) ALT (alternative lengthening of telomeres) is involved in supporting cell proliferation in tailfin regeneration. The experimental system employs a quantitative area-based measurement as a measure of the degree of regeneration. Functional studies used antisense morpholino gene knockdown and chemical inhibition to implicate ALT involvement.

****Major comments:****

The experimental logic is appropriate, and in general, the data support the conclusions. Strengths of the work include: (1) The quantitative measure of % regeneration appears to be quite objective; (2) the internally controlled experimental design of the morpholino knockdown experiments of Fig 5.

We thank the reviewer for the comment

The Western blot in Fig S2 has some issues. The image is a montage. The experiment appears to have been done only once. The band's identifications by kDa are imprecise (where is the 82 kDa band on the gel? - there are bands smaller and larger than 82 kDa, but none of 82kDa; the 50 kDa band is close to background; the DMSO lane is underloaded relative to the two test lanes (but as the observation is a reduction in the test samples, this does not result in a misinterpretation). What concentration of ATRinhIV were used? The blot has 10 and 50 microM, Fig S1B has 50 and 100 microM, and the text says 1-50 microM).

We have performed new regeneration experiments in wild type adult fish using ATR inhibitor. The results show that treating fish with ATR inhibitor results in a clear decrease in the overall phosphorylation status of ATR/ATR substrates within the regenerated tissue (Figure 5B and C). In this case, the intensity of the whole lane was used for quantification. As mentioned in the text, we used concentrations of 1, 10 and 50 microM, but we do not observe any difference with the 1

microM concentration, thus do not show it. Then we measured the regeneration capacity in both wild type and tert mutant fish using 10microM concentration

The MO-knockdown studies are interpreted as showing synergy of atr and tert knockdown.

There are two problems with their interpretation of synergy: (1) the single result of a greater effect with both MOs does not distinguish between an additive or synergistic effect (and synergistic action is by definition a greater than additive action);

We agree with the reviewer's comment, and have removed the sentence "Interestingly a synergistic effect was observed when both mechanisms are inhibited" from the Results section.

(2) MO dose is not controlled by a group with an equal total MO doses (mo-std+mo-atr and mo-std+mo-tert). While acknowledging that the issues of using local MO delivery in an adult model are very different from global delivery in an embryonic model, the "synergy" interpretation still requires these experiments/controls be done. These experiments were not accompanied by any molecular evidence that either of the morpholinos targeted expression of the intended gene (which would likely have to be derived from their assessment in another system) - a control that can be challenging, but one that is regarded as essential in the field (<https://doi.org/10.1242/dev.001115>). While this will be difficult to do in the adult setting, it is still appropriate to validate the activity/molecular efficacy of the MO sequence in an experimentally tractable scenario. The specificity of this experiment and interpretation would also be enhanced and corroborated independently by undertaking the atr knockdown in the tert -/- mutant background. Overall, these experiments were preliminary and require further work that could be done within 3 months.

We have performed regeneration experiments using WT fish to address this issue. We analyzed the regenerated area of control and morpholino injected fish and then obtained regenerating blastema and analyzed the expression of *tert* and *atr*. The results are shown in Supplemental Figure S4 (A-E). The regeneration capacity is inhibited in tissues injected with a mix of mo-std+mo-ter, a mix of mo-std+mo-atr, or a mix of mo-tert+mo-atr compared with a control injected with a double dose of mo-std (std 2x, Fig S4B). In addition, the expression of *tert* and *atr* is decreased in the regenerated blastema upon morpholino injection (Fig S4 C and D) indicating that the genetic inhibition of the expression of these genes was efficient. Finally, the levels of TERRA RNAs are increased upon amputation and this induction is reduced when we mo-atr or a combination of mo-atr+tert were microinjected (Fig S4E).

We have also performed caudal fin regeneration experiments in tert mutant fish microinjected with mo-atr and mo-nbs1, and analyzed the levels of TERRA RNAs and C-circles amount. The results are shown in Supplemental Figure S4. As expected, regeneration capacity of the caudal fins of fish microinjected with both morpholinos decreased compared to control fish (Fig S4 F-H). Consistently, TERRA RNAs levels, as well as C-circles amount, increased in the regenerating tissue and this induction was lower when atr and nbs1 gene expression was decreased by mo-injection (Fig S4 I and J). Taking altogether, these results indicate that ALT mechanism is induced upon an injury and operating in the regenerating tissue of both wild type and tert deficient fish.

Note - the tert MO sequence is missing from the table in Fig S4.

The sequence has been added

The adult experiments have used n=6-10 animals/group. There is no consideration of statistical power (is the analysis of Fig 1C adequately powered?).

The type of statistical test applied in Fig 1C (2-way ANOVA, plus Dunnett's post-test) compares means of every clip among the 3 genotypes. This is the test that is recommended for this kind of data and experiment.

The degree and nature of replication is not clear in all cases. For example, in Fig 1, were the 6 fish run as one cohort of 6 animals in parallel (which would be just one experiment with 6 animals, each animal being a biological replicate), or were there 6 animals injured at different times (representing multiple independent experiments and represented a greater degree of reproducibility), or something in between. A similar question applies to the other figures.

In the experiments, 6 fish total were used per group sampled in at least 2 independent experiments. This has been included in the figure legend and in the Mat&Met section

For the experiment of Fig 6F, although there are ≥ 100 larvae per group, it is not clear that this experiment has been done more than once.

In the conducted experiments, three independent trials were conducted. The total number of larvae per group utilized in each of the three distinct experiments surpassed 100 larvae per group (approximately 40 larvae in each independent experiment). This data has been incorporated into both the figure legend and the Materials and Methods section."

A few comments about data presentation. "Regeneration rate" and its derivatives are presented as mean \pm SEM. The parameter measured is correctly defined in methods as "Percent fin regeneration", however the graphs where it is plotted have the y-axis labelled as "regeneration rate (%)" (for example, Fig 1B), which is incorrect. The plotted parameter is not a rate - although there is a time dimension (x-axis), what is plotted at each time point is "% regeneration".

This has been corrected and y-axis is now labeled as Regeneration area (% of initial fin area).

Also, in most figures, such as Fig 1B and 1C, mean \pm SD would be more appropriate, as here each of the $n=6$ data points represents a single observation from one individual in the population, not the mean of 6 small samples of groups of individuals from the population. Furthermore, at these small n -values (6-10 through the report), scatter plots are considered a more appropriate way of displaying the data (some succinct references: DOI: 10.4103/2229-3485.100662 ; from a Nature group journal DOI: 10.1038/s41551-017-0079 ; from a PLOS journal <https://doi.org/10.1371/journal.pbio.1002128>).

This was a mistake in the figure legend, since Fig 1B was already showing mean \pm SD. Fig 1C is now showing mean \pm SD and has been represented with scatter plots.

The use of mean \pm SEM in Fig 4 could be appropriate, but as n is "at least two independent experiments" scatter plots would again be appropriate. Readers would then know which data sets had only two values.

In two instances, the same data are presented in two different ways (Fig 1B, 1C; the column graphs and arrows of Fig 3D).

Fig4 is presented now as scatter plot graphs

How does "data are average of at least 2 independent experiments" apply to Fig 3C?

In the experiments in Fig3C, "Data are average of 2 independent experiments of 3 fish per group pooled". This has been included in the figure legend.

****Minor comments:****

The paper is written clearly overall. There are multiple minor grammatical/typographical errors, but these did not detract from understanding the manuscript. These were most abundant in the discussion.

A few points:

Discussion p1 - what is meant by "prematurely aged 11-month fish"

This refers to 11 months old *tert*^{-/-} fish, which has been shown to present accelerated aging features at this age compared to wild type

Discussion p2 - you mean "doubled" rather than duplicated?

Yes; this has been corrected in the new version

tert^{+/+}, *tert*^{+/-} and *tert*^{-/-} genotypes for experiments - how were these obtained and genotypically verified? (heterozygous incrosses? WT x homozygous mutant outcrosses?)

All the fish adult fish of the 3 genotypes were obtained from heterozygous incrosses. Then fish were genotyped by PCR. A protocol for genotyping has been added in the Mat&Met section. The wild type larvae used in the tail fin regeneration experiments inhibiting ATR were obtained by wild type cross, whereas the *tert*^{-/-} were obtained by *tert*^{-/-} incross

The last paragraph of the discussion makes some valid points, but it seemed out of place and I wondered if it was misplaced.

This paragraph is added to highlight that our work describes new in vivo model to perform drug screening to inhibit ALT mechanism of telomere maintenance, which is of particular importance for the survival of ALT positive tumor cells.

The *rps11* primers appear in the Table of Fig S3 twice.

This has been corrected

Reviewer #3 (Significance (Required)):

The authors claim that this is the first in vivo model examining ALT in regeneration.

The paper contributes to the relatively small body of literature using adult zebrafish models (rather than embryonic larval models) in biomedical research. I cannot comment on the telomere/telomerase literature.

This report will be of interest to those working in regenerative medicine, telomere biology, cancer research, and those interested in zebrafish models of disease and physiological processes.

My expertise encompasses zebrafish disease models and functional studies; I do not have special expertise in telomerase or ALT pathways.

We fully appreciate the reviewer's comments!

Dear Dr. Cayuela,

Thank you for the transfer of your revised manuscript from Review Commons to our editorial offices, that had also been re-reviewed before at one of the partner journals. I have now received the report from the referee that I asked to evaluate your further revised study (original referee #1), which can be found at the end of this email. Original referee #2 was already satisfied with the previous version, whereas referee #3 declined to look at the revised previous version and was therefore not contacted again.

As you will see, the referee in principle supports publication of your study in EMBO reports but has remaining concerns and suggestions to improve the manuscript, I ask you to address in a final revised manuscript doing the requested text changes. Please also provide a final p-b-presence to the remaining referee points and the editorial requests (see below).

Moreover, the manuscript now also needs formatting according to our journal style. Please carefully review the instructions that follow below.

- 1) a .docx formatted version of the final manuscript text (including legends for main figures, EV figures and tables), but without the figures included. Figure legends should be compiled at the end of the manuscript text.
- 2) individual production quality figure files as .eps, .tif, .jpg (one file per figure), of main figures and EV figures. Please upload these as separate, individual files upon re-submission.

For more details, please refer to our guide to authors:
<http://www.embopress.org/page/journal/14693178/authorguide#manuscriptpreparation>

Please consult our guide for figure preparation:
http://wol-prod-cdn.literatumonline.com/pb-assets/embo-site/EMBOPress_Figure_Guidelines_061115-1561436025777.pdf

See also the guidelines for figure legend preparation:
<https://www.embopress.org/page/journal/14693178/authorguide#figureformat>

- 3) a .docx formatted letter INCLUDING the reviewers' report and your detailed point-by-point response to the comments. As part of the EMBO Press transparent editorial process, the point-by-point response is part of the Review Process File (RPF), which will be published alongside your paper.

- 4) a complete author checklist, which you can download from our author guidelines (<https://www.embopress.org/page/journal/14693178/authorguide>). Please insert page numbers in the checklist to indicate where the requested information can be found in the manuscript. The completed author checklist will also be part of the RPF.

Please also follow our guidelines for the use of living organisms, and the respective reporting guidelines:
<http://www.embopress.org/page/journal/14693178/authorguide#livingorganisms>

- 5) that primary datasets produced in this study (e.g. RNA-seq, ChIP-seq, structural and array data) are deposited in an appropriate public database. If no primary datasets have been deposited, please also state this in a dedicated section (e.g. 'No primary datasets have been generated and deposited'), see below.

The accession numbers and database should be listed in a formal "Data Availability" section that follows the model below. This is now mandatory (like the COI statement). Please note that the Data Availability Section is restricted to new primary data that are part of this study. This section is mandatory. As indicated above, if no primary datasets have been deposited, please state this in this section

Data availability

6) We now request the publication of original source data with the aim of making primary data more accessible and transparent to the reader. You will receive a separate email with instructions for providing source data with your revised manuscript, including information how to upload and organize the files.

8) Regarding data quantification and statistics, please make sure that the number "n" for how many independent experiments were performed, their nature (biological versus technical replicates), the bars and error bars (e.g. SEM, SD) and the test used to calculate p-values is indicated in the respective figure legends (also for EV and Appendix figures). Please also check that all the p-values are explained in the legend, and that these fit to those shown in the figure. Please provide statistical testing where applicable. Please avoid the phrase 'independent experiment', but clearly state if these were biological or technical replicates. Please also indicate (e.g. with n.s.) if testing was performed, but the differences are not significant. In case n=2, please show the data as separate datapoints without error bars and statistics. See also: <http://www.embopress.org/page/journal/14693178/authorguide#statisticalanalysis>

9) Please add scale bars of similar style and thickness to microscopic images, using clearly visible black or white bars (depending on the background). Please place these in the lower right corner of the images themselves. Please do not write on or near the bars in the image but define the size in the respective figure legend.

10) Please also note our reference format:

12) We now use CRediT to specify the contributions of each author in the journal submission system. CRediT replaces the author contribution section. Please use the free text box to provide more detailed descriptions and do NOT provide your final manuscript text file with an author contributions section. See also our guide to authors: <https://www.embopress.org/page/journal/14693178/authorguide#authorshippinguidelines>

13) All Materials and Methods need to be described in the main text using our 'Structured Methods' format, which is required for all research articles. According to this format, the Methods section should include a Reagents and Tools Table (listing key reagents, experimental models, software, and relevant equipment and including their sources and relevant identifiers), uploaded as separate file, and a Methods section in which we encourage the authors to describe their methods using a step-by-step protocol format with bullet points, to facilitate the adoption of the methodologies across labs. More information on how to adhere to this format as well as downloadable templates (.doc) for the Reagents and Tools Table can be found in our author guidelines

(section 'Structured Methods'):

14) Please add 5 keywords to the manuscript and order the manuscript sections like this, using these names:

Title page - Abstract (max. 175 words) - Keywords - Introduction - Results - Discussion - Methods - Data availability section - Acknowledgements - Disclosure and Competing Interests Statement - References - Figure legends - Expanded View Figure legends

15) Please make sure that all the funding information is also entered into the online submission system and that it is complete and similar to the one in the acknowledgement section of the manuscript text file.

Moreover, please note that all corresponding authors are required to supply an ORCID ID for their name upon submission of a revised manuscript. Please find instructions on how to link the ORCID ID to the account in our manuscript tracking system in our Author guidelines: <http://www.embopress.org/page/journal/14693178/authorguide#authorshipguidelines>

I look forward to seeing your revised manuscript when it is ready.

Best,

Referee #1:

Going through the point-by-point responses by the authors, I feel that the manuscript has increased in clarity and quality. Nonetheless, I would suggest that the authors use their responses to further explain unclear key results in the discussion section. We understand their scientific reasoning but the manuscript would benefit from including short versions of these explanations in the discussion, namely for non-experts.

The explanations I think should be discussed are the following:

1.
REVIEWER: "Although the authors could provide more evidence for ALT involvement as shown in Fig S4, how do they explain that TERRA levels are brought to base levels (T0) while c-circles, although decreased, remain higher than at T0?"

AUTHORS: "This can likely be explained by the inherent differences in the stability of these molecules. It is well-established that circular extrachromosomal DNA, such as c-circles, is structurally more stable than linear DNA and even more so than RNA. This exceptional stability has led to the investigation of circular DNA as potential biomarkers in various pathologies. In contrast, TERRA RNA is subject to more dynamic regulation and degradation, likely mediated by various regulatory proteins (PMID: 37107548, PMID: 37060569). Therefore, it can be speculated that while c-circles remain relatively stable due to their structural properties, TERRA RNA is under stricter control, leading to its faster normalization during tissue regeneration. Additionally, it is possible that normalization of c-circle levels would require measurements beyond the 48-hour time point. These distinct regulatory and stability profiles likely explain the differential behavior observed in level normalization in our experiments."

2.
REVIEWER: "Why does not telomere length change in the morpholino injection conditions?"

AUTHORS: "The lack of change in telomere length observed under morpholino injection conditions may be attributed to the

activation of alternative telomere maintenance mechanisms that remain unaffected by the morpholinos used. Specifically, while the morpholino targeting the nbs1 and atr genes effectively disrupts the canonical alternative lengthening of telomeres (ALT) pathway, the morpholino targeting tert inhibits telomerase-mediated telomere lengthening. Consequently, the inhibition of one mechanism may compensate for the other, allowing for the maintenance of telomere length during the regeneration process. This complexity highlights the interplay of multiple pathways involved in sustaining telomere length under these experimental conditions."

3.

REVIEWER: "The authors mention ATRX and DAXX act as ALT suppressors in presence of a mutation, while if there is no ATRX and DAXX, ALT is activated. What about when ATRX and DAXX are present but without the mutation? What is this mutation? Do their gene expression analysis take into consideration this modification?"

AUTHORS: "ATRX and DAXX are ALT suppressors and their loss of function mutations lead to the induction of ALT process. For instance, the ALT positive osteosarcoma cell line U2-OS presents a homozygous deletion of exons 2 to 19 in ATRX gene that provokes its loss of expression (PMID: 21719641). ATRX and DAXX genes in wt and tert^{-/-} fish do not present any mutation. However, the caudal fin amputation induces a clear down-regulation of both ATRX and DAXX gene expression in tert^{-/-}, but not in wild type, fish (Fig 4C and D), and this could be triggering ALT in regenerating caudal fins"

4.

REVIEWER: "In line 53, it is missing that atrx and daxx expression was found decreased in tert mutants but not the WT. This should also be discussed further, why is it?"

AUTHORS: "This observation raises an intriguing question about the potential role of telomerase in maintaining the expression of these genes. The precise mechanisms underlying this relationship remain unclear. Further experimentation would be required to elucidate the functional implications of this finding and the potential regulatory pathways involved. We acknowledge that this is a valuable question that warrants additional investigation, and we will continue to explore the roles of tert and ALT in the context of regeneration."

5.

REVIEWER: "The strongest argument would be the lack of changes in telomere length. However, it is unclear to me whether in this model there are sufficient cell divisions to observe telomere shortening. Indeed, a few divisions in stem cells may be sufficient to ensure the regeneration of the fin, thus not affecting the total telomere length even in the absence of TERT. I would like to see an additional assay, and I would prefer to see the primary data rather than quantification."

AUTHORS: "We could agree with the reviewer's comment, and it could be possible that a few divisions may be sufficient for ensure complete regeneration of caudal fin. However the experiments performed with tert mutants involving 12 consecutive cycles of fin amputation and regeneration (7 days each cycle) show that tert^{-/-} fish are capable of regenerate caudal fin at a similar level than wild type fish do (Fig 2B and C). Furthermore, they also maintain similar telomere length (Fig 2D and E). We do think that 84 days (12x7) of continuous cell divisions would result in telomere length attrition and impaired regeneration capacity; unless a tert independent mechanism of telomere lengthening is operating."

6.

REVIEWER: "The downregulation of ATRX is very puzzling and would require additional data to be explained. In ALT-positive cancers, ATRX is mutated, and I am not sure I understand why ATRX would be downregulated in this setting."

AUTHORS: "How it has been responded to reviewer #2, ATRX and DAXX are ALT suppressors and their loss of function mutations lead to the induction of ALT process. For instance, the ALT positive osteosarcoma cell line U2-OS presents a homozygous deletion of exons 2 to 19 in ATRX gene that provokes its loss of expression (PMID: 21719641). ATRX and DAXX genes in wt and tert^{-/-} fish do not present any mutation. However, the caudal fin amputation induces a clear down-regulation of both ATRX and DAXX gene expression in tert^{-/-}, but not in wild type, fish (Fig 4C and D), and this could be triggering ALT in regenerating caudal fins. To demonstrate that the regeneration process in tert^{-/-} fish is driven by ALT and depends on the downregulation of atrx, we conducted experiments to overexpress the zebrafish atrx gene in the regenerating blastema of tert mutants. We microinjected a plasmid construct expressing atrx under the control of a CMV promoter into the blastema of tert^{-/-} fish. Notably, we observed that caudal fin regeneration in these fish was significantly impaired upon atrx overexpression compared to control-injected fish. This finding strongly suggests that the ALT mechanism is operational in the regeneration of tert^{-/-} caudal fins and that its effectiveness is at least partially dependent on atrx downregulation. The mechanisms behind the downregulation of atrx are indeed compelling and warrant deeper experimentation. One could speculate that absence of telomerase activity may promote the downregulation of atrx. Such hypotheses, among others, may serve as valuable subjects for future research into fin regeneration, which we hope to explore in light of the results from this paper."

Rev_Com_number: RC-2021-01152

New_manu_number: EMBOR-2025-62098V1-T

Corr_author: Cayuela

Title: Telomerase and ALT coexist in the regenerating zebrafish caudal fins

Referee #1:

Going through the point-by-point responses by the authors, I feel that the manuscript has increased in clarity and quality. Nonetheless, I would suggest that the authors use their responses to further explain unclear key results in the discussion section. We understand their scientific reasoning but the manuscript would benefit from including short versions of these explanations in the discussion, namely for non-experts.

We fully appreciate the reviewer's comments on the manuscript, and also thank the efforts to improve it.

The explanations I think should be discussed are the following:

1.

REVIEWER: "Although the authors could provide more evidence for ALT involvement as shown in Fig S4, how do they explain that TERRA levels are brought to base levels (T0) while c-circles, although decreased, remain higher than at T0?"

AUTHORS: "This can likely be explained by the inherent differences in the stability of these molecules. It is well-established that circular extrachromosomal DNA, such as c-circles, is structurally more stable than linear DNA and even more so than RNA. This exceptional stability has led to the investigation of circular DNA as potential biomarkers in various pathologies. In contrast, TERRA RNA is subject to more dynamic regulation and degradation, likely mediated by various regulatory proteins (PMID: 37107548, PMID: 37060569). Therefore, it can be speculated that while c-circles remain relatively stable due to their structural properties, TERRA RNA is under stricter control, leading to its faster normalization during tissue regeneration. Additionally, it is possible that normalization of c-circle levels would require measurements beyond the 48-hour time point. These distinct regulatory and stability profiles likely explain the differential behavior observed in level normalization in our experiments."

This is discussed in lines 350-356

2.

REVIEWER: "Why does not telomere length change in the morpholino injection conditions?"

AUTHORS: "The lack of change in telomere length observed under morpholino injection conditions may be attributed to the activation of alternative telomere maintenance mechanisms that remain unaffected by the morpholinos used. Specifically, while the morpholino targeting the nbs1 and atr genes effectively disrupts the canonical alternative lengthening of telomeres (ALT) pathway, the morpholino targeting tert inhibits telomerase-mediated telomere lengthening. Consequently, the inhibition of one mechanism may compensate for the other, allowing for the maintenance of telomere length during the regeneration process. This complexity highlights the interplay of multiple pathways involved in sustaining telomere length under these experimental conditions."

This is discussed in lines 362-372

3.

REVIEWER: "The authors mention ATRX and DAXX act as ALT suppressors in presence of a mutation, while if there is no ATRX and DAXX, ALT is activated. What about when ATRX and DAXX are present but without the mutation? What is this

mutation? Do their gene expression analysis take into consideration this modification?"

AUTHORS: "ATRX and DAXX are ALT suppressors and their loss of function mutations lead to the induction of ALT process. For instance, the ALT positive osteosarcoma cell line U2-OS presents a homozygous deletion of exons 2 to 19 in ATRX gene that provokes its loss of expression (PMID: 21719641). ATRX and DAXX genes in wt and tert^{-/-} fish do not present any mutation. However, the caudal fin amputation induces a clear down-regulation of both ATRX and DAXX gene expression in tert^{-/-}, but not in wild type, fish (Fig 4C and D), and this could be triggering ALT in regenerating caudal fins"

This is discussed in lines 332-342

4.

REVIEWER: "In line 53, it is missing that atrx and daxx expression was found decreased in tert mutants but not the WT. This should also be discussed further, why is it?"

AUTHORS: "This observation raises an intriguing question about the potential role of telomerase in maintaining the expression of these genes. The precise mechanisms underlying this relationship remain unclear. Further experimentation would be required to elucidate the functional implications of this finding and the potential regulatory pathways involved. We acknowledge that this is a valuable question that warrants additional investigation, and we will continue to explore the roles of tert and ALT in the context of regeneration."

This is discussed in lines 332-342

5.

REVIEWER: "The strongest argument would be the lack of changes in telomere length. However, it is unclear to me whether in this model there are sufficient cell divisions to observe telomere shortening. Indeed, a few divisions in stem cells may be sufficient to ensure the regeneration of the fin, thus not affecting the total telomere length even in the absence of TERT. I would like to see an additional assay, and I would prefer to see the primary data rather than quantification."

AUTHORS: "We could agree with the reviewer's comment, and it could be possible that a few divisions may be sufficient to ensure complete regeneration of caudal fin. However the experiments performed with tert mutants involving 12 consecutive cycles of fin amputation and regeneration (7 days each cycle) show that tert^{-/-} fish are capable of regenerate caudal fin at a similar level than wild type fish do (Fig 2B and C). Furthermore, they also maintain similar telomere length (Fig 2D and E). We do think that 84 days (12x7) of continuous cell divisions would result in telomere length attrition and impaired regeneration capacity; unless a tert independent mechanism of telomere lengthening is operating."

This is discussed in lines 288-303

6.

REVIEWER: "The downregulation of ATRX is very puzzling and would require additional data to be explained. In ALT-positive cancers, ATRX is mutated, and I am not sure I understand why ATRX would be downregulated in this setting."

AUTHORS: "How it has been responded to reviewer #2, ATRX and DAXX are ALT

suppressors and their loss of function mutations lead to the induction of ALT process. For instance, the ALT positive osteosarcoma cell line U2-OS presents a homozygous deletion of exons 2 to 19 in ATRX gene that provokes its loss of expression (PMID: 21719641). ATRX and DAXX genes in wt and *tert*^{-/-} fish do not present any mutation. However, the caudal fin amputation induces a clear down-regulation of both ATRX and DAXX gene expression in *tert*^{-/-}, but not in wild type, fish (Fig 4C and D), and this could be triggering ALT in regenerating caudal fins. To demonstrate that the regeneration process in *tert*^{-/-} fish is driven by ALT and depends on the downregulation of *atrx*, we conducted experiments to overexpress the zebrafish *atrx* gene in the regenerating blastema of *tert* mutants. We microinjected a plasmid construct expressing *atrx* under the control of a CMV promoter into the blastema of *tert*^{-/-} fish. Notably, we observed that caudal fin regeneration in these fish was significantly impaired upon *atrx* overexpression compared to control-injected fish. This finding strongly suggests that the ALT mechanism is operational in the regeneration of *tert*^{-/-} caudal fins and that its effectiveness is at least partially dependent on *atrx* downregulation. The mechanisms behind the downregulation of *atrx* are indeed compelling and warrant deeper experimentation. One could speculate that absence of telomerase activity may promote the downregulation of *atrx*. Such hypotheses, among others, may serve as valuable subjects for future research into fin regeneration, which we hope to explore in light of the results from this paper."

This is discussed in lines 332-342

Dear Dr. Cayuela,

Thank you for the submission of your revised manuscript, and I am sorry for the delay in getting back to you. I have taken over the handling of your ms from my colleague Achim who is currently not in the office.

The files look good overall, but a few more editorial requests will need to be addressed. I am making another decision now so that you can easily replace files. You should be able to bring forward all files from the last version of your ms and then you can replace what needs to be replaced. If you have any questions regarding the process please write to contact@emboreports.org.

- There is one author name discrepancy: Monique Anchemin-Flageul in the ms vs. Monique Anchemin in our online submission system, please correct.
- DATA NOT SHOWN on page 7 is not allowed per journal policy. Please either show the data or re-phrase.
- The ORCID is still missing for García-Castillo - it was requested on the 22.7.25 but it seems to be still missing.
- The FUNDING INFO is missing some information in our online submission system: European Union, Fundación Ramón Areces; and other info is missing in the ms: "ZEBER" (by Consejería de Sanidad de la Región de Murcia). All funding information must be present in both places, please correct.
- There are figure callouts for individual panels of Fig. S4 - these need to be updated as you only have EV figures.
- Both tables in the Appendix file should be part of the Reagents and Tools table. Please transfer this information and delete the Appendix file. The callouts "Appendix Information Table" also need to be deleted/corrected.
- The synopsis image you sent is good but we still need the synopsis text for our website: A) a short (1-2 sentences) summary of the findings and their significance, B) 2-3 bullet points highlighting key results.
- on page 12 - [dx.doi.org/10.17504/protocols.io.mrjc54n](https://doi.org/10.17504/protocols.io.mrjc54n) is only in the ms text, but not listed in the references, please add this to the reference list.
- The source data (SD) need to be uploaded as 1 folder per main figure. The SD for EV figures can all be in one folder.
- The email for co-author Monique Anchemin - monicanchemin@hotmail.com bounced. You either need to remove the author from the author list in the system and add her using a new email address or send us the new email address and we will update the author's account accordingly. All authors need to have functional email addresses.

Figure Legends - Comments

- Please note that the exact p values are not provided in the legends of figures 1B, C; 2C, E; 3A, C, D; 4A, B, C, D, F, G; 5C, E; 6C-E; EV1 A, EV2 A, EV4 B-J; EV5 C. Please provide exact p-values as reasonable.

I would like to suggest some minor changes to the title and abstract that needs to be written in present tense. Please let me know whether you agree with this:

Telomerase and Alternative Lengthening of Telomeres coexist in the regenerating zebrafish fin

Telomeres are essential for chromosome protection and genomic stability, and telomerase function is critical for organ homeostasis. Zebrafish has become a useful vertebrate model for understanding cellular and molecular mechanisms of regeneration. The regeneration capacity of the caudal fin of wild-type zebrafish is not affected by repetitive amputation, but the behavior of telomeres during this process has not yet been studied. Here, we characterize the regeneration process in a telomerase-deficient zebrafish model, and study the regenerative capacity after repetitive amputations at different ages. We show that the regenerative efficiency decreases with aging in all genotypes but telomere length is maintained even in telomerase-deficient fish. Our data indicate that telomere length can be maintained by the regenerating cells through the recombination-mediated Alternative Lengthening of Telomeres (ALT) pathway, which likely supports high rates of cell proliferation during the caudal fin regeneration process.

All editorial and formatting issues were resolved by the authors.

Dr. Maria L. Cayuela
IMIB-Pascual Parrilla
Telomerase, Cancer and Aging. Surgery Department
Murcia
Spain

Dear Dr. Cayuela,

I am very pleased to accept your manuscript for publication in the next available issue of EMBO reports. Thank you for your contribution to our journal.

Yours sincerely,
